# Adaptative machine vision with microsecond-level accurate perception beyond human retina

Ling Li [1], Shasha Li[2], Wenhai Wang[1], Jielian Zhang[1], Yiming Sun[1], Qunrui Deng[1], Tao Zheng[1], Jianting Lu[3], Wei Gao[1], Mengmeng Yang[1], Hanyu Wang[1], Yuan Pan[1], Xueting Liu[1], Yani Yang[1], Jingbo Li[4,5] & Nengjie Huo [1,5] ✉

Visual adaptive devices have potential to simplify circuits and algorithms in machine vision systems to adapt and perceive images with varying brightness levels, which is however limited by sluggish adaptation process. Here, the avalanche tuning as feedforward inhibition in bionic two-dimensional (2D) transistor is proposed for fast and high-frequency visual adaptation behavior with microsecond-level accurate perception, the adaptation speed is over $10^4$ times faster than that of human retina and reported bionic sensors. As light intensity changes, the bionic transistor spontaneously switches between avalanche and photoconductive effect, varying responsivity in both magnitude and sign (from $7.6 \times 10^4$ to $-1 \times 10^3$ A/W), thereby achieving ultra-fast scotopic and photopic adaptation process of 108 and 268 μs, respectively. By further combining convolutional neural networks with avalanche-tuned bionic transistor, an adaptative machine vision is achieved with remarkable microsecond-level rapid adaptation capabilities and robust image recognition with over 98% precision in both dim and bright conditions.

In the human retina, the photoreceptor (rod and cone cells) first transmits light stimulation to the horizontal and bipolar neurons, where the brain receives signals through a feedforward excitation circuit formed with bipolar neurons, and the horizontal neurons output inhibitory signals to receptors through feedback inhibitory circuit depending on the magnitude of the light stimulus. Notably, due to the higher photosensitivity of rod than cone cells by three orders, the rod cells activate the primary photoreceptor in weak light, and the inhibitory signal serves as a selector switching the primary photoreceptor to cone cells when shifting to strong light[1,2]. This spontaneous visual behavior is known as visual adaptation[3,4], because of which, the retina prevents the brain from continuously receiving overstimulated information when the environmental brightness changes excessively and

rapidly. However, this visual behavior relies on horizontal neurons modulated through feedback inhibitory with significant time hysteresis, which introduces serious hazards in daily life, such as car accidents, blindness and high difficulty in searching at night[5–7]. Therefore, it is crucial to optimize the visual behavior, the crucial aspect of which is to optimize the feedback inhibition circuit for fast and high-frequency visual adaptation.

Machine vision technology, built on deep learning with convolutional neural networks, achieves highly accurate image recognition. This breakthrough technology demonstrates immense potential in areas such as autonomous driving, facial recognition, and medical imaging, that can replace the human retina in hazardous environments to perceive and judge[8–11]. However, the currently developed vision

[1]School of Semiconductor Science and Technology, South China Normal University, Foshan 528225, P.R. China. [2]School of Electronic Engineering, Chaohu University, Hefei 238000, China. [3]National Key Laboratory of Science and Technology on Reliability Physics and Application of Electronic Component, China Electronic Product Reliability and Environmental Testing Research Institute, Guangzhou 510610, China. [4]College of Optical Science and Engineering, Zhejiang University Hangzhou 310027, China. [5]Guangdong Provincial Key Laboratory of Chip and Integration Technology, Guangzhou 510631, P.R. China. ✉e-mail: njhuo@m.scnu.edu.cn

perception systems struggle to adapt to varying brightness levels because of the impressionable image quality by brightness conditions, thus requiring complex circuits and advanced algorithms in machine vision. Recently, Prof. Chai Yang's team proposed an innovative concept that can significantly reduce the requirements of circuits and algorithms by combining a two-dimensional (2D) bionic vision sensor with visual adaptation capabilities and a convolutional neural network[12]. It is demonstrated that the ultrathin 2D materials are appropriate for developing bionic visual sensors due to excellent electrostatic doping, fast photoresponse, high avalanche performance and ferroelectric properties[12–18]. Although, 2D bionic vision sensors have been well developed with bio-inspired photopic and scotopic adaptation, but still encounter some issues or disadvantages, limiting their combination to machine vision. For example, the bilayer $MoS_2$ transistor with surface trap states exhibits a visual adaptation function enabled by the charge trapping and detrapping mechanism under gate voltage modulation, that takes up to several minutes and requires manual configuration of different gates under both photopic and scotopic adaptation conditions[4]. The γ-InSe device can only simulate the photopic adaptation function through photo-pyroelectric and photo-thermoelectric effects[5]. The $ReS_2$/U6N bionic device utilizes the interfacial defects as feedback inhibition to induce visual behavior at limited wavelength also with prolonged adaptation time[7]. Therefore, the defects tuning as feedback inhibition in commonly reported 2D bionic visual device always leads to the slow adaption process just as retina behaves. The visual bionic mechanism needs to be explored to further optimize the visual adaptation that is beyond the human retina and the cutting-edge machine vision systems.

To solve the above issues and introduce a distinctive working mechanism for visual adaptation, we developed a bionic visual 2D transistor. A significant avalanche effect, which can be tuned by external voltage ($V_{DS}$ and $V_{GS}$) and light illumination conditions, is enabled by the impact ionization in the depletion region formed in the ultrathin $MoS_2$ channel with top $WSe_2$. At $V_{GS} = -3\,V$, the breakdown voltage ($V_{EB}$) is as low as 5.48 V and the multiplication factor reaches $5.29 \times 10^5$, superior to the reported 2D avalanche transistors. By changing the environmental illumination from dim to bright, the output current increases and then decreases, showing retinal visual behavior. This is caused by the spontaneous transition of the device's operating mechanism from a high-sensitivity avalanche effect to a low-sensitivity photoconductivity effect, whose function is like a switch-over between rod and cone cells in the retina upon the environment change. Taking advantage of avalanche tuning operation as a feed-forward inhibition circuit in a bionic neural network, the device can emulate high-frequency visual behavior at 6 and 3 kHz under simulated scotopic and photopic adaptation conditions, possessing a fast adaptation process of 108 and 268 μs, respectively, that is far beyond human retina function and the currently developed 2D bionic sensors. The −3 dB bandwidth reaches 10.5 kHz at a weak light power of 125.62 pW due to the defect filling by the carrier avalanche, that is also surpassing the dynamic response of the retina (500 Hz). Leveraging the microsecond-level rapid adaptation capability, the bionic avalanche transistor is further seamlessly combined with a convolutional neural network, developing an ultra-fast adaptive machine vision system, which excels in image recognition in both dim and bright environmental conditions, boasting a precision rate exceeding 98%.

## Results

### Retina and neural circuit motifs

As shown in Fig. 1a, the human retinal architecture consists of rich visual cells whose autonomic processing of light information is known as visual adaptation[19]. The types of neurons and the connection paths can form a variety of circuit motifs that are key to visual adaptation behavior. Figure 1b shows three typical examples in the visual system.

Light information is transmitted from the retina to the cerebral cortex via feedforward excitation, which enables information convergence and divergence[20]. Convergence improves the signal-to-noise ratio and divergence allows the signal to be processed by multiple channels to enhance computational performance. The feedback inhibition is an important circuit for visual adaptation, that involves the regeneration/bleaching photopigment and causes a long-time adaptation process[21,22]. Feedforward inhibition exists in the cerebral cortex that is faster and more predictive than feedback inhibition[23]. In this circuit, the target cell receives an integration of excitatory and inhibitory information, effectively avoiding the long adaptation process. However, almost all reported bionic visual systems adopted a feedback inhibition such as charge trapping or detrapping process to emulate the visual adaption, while the faster feedforward inhibition has never been employed in bio-inspired devices so far.

Figure 1c, d show the photopic/scotopic adaptation process of the bionic devices relying on feedback inhibition and feedforward inhibition circuit, respectively. In the former case, the visual adaptation involves an overexcited signal change at the primary stage (Process 1) by the application of extra gate voltages ($V_{GS}$), and a long-time adaptation process with a timescale of minutes (Process 2) caused by the prolonged charge trapping/detrapping process. As a more advantageous situation, the feedforward inhibitory circuit-based visual adaptation avoids receiving overexcited signals and proceeds much faster with a timescale of microseconds. In addition, the range of current variation is larger, indicating a more obvious excitation (inhibition) effect. To be different from previous bionic devices based on feedback inhibition, this work will employ a faster and more predictive feedforward inhibition in an avalanche transistor for ultra-fast and high-frequency visual adaptation behavior.

### Device scheme and characterization

Figure 2a and Supplementary Fig. 1 illustrate the schematic diagram and optical microscopy image of the device structure and electrical connections, respectively. This junction field effect transistor (JFET) consists of an ultrathin $MoS_2$ transport channel and top $WSe_2$ gate, where the depletion region and vertical electric field at the $MoS_2$/$WSe_2$ interface can be modulated by top gate voltages to control the switching behavior and avalanche effect. The detailed device fabrication process is presented in the Methods. The cross-sectional high-resolution transmission electron microscopy image is shown in Fig. 2b, demonstrating the clear lattice fringe and ultrathin thickness of $MoS_2$ (3.93 nm) and $WSe_2$ (3.04 nm), ensuring a smooth and clean van der Waals (vdW) interface. To further identify the composition of the stacking layers, Fig. 2c and Supplementary Fig. 2 show the energy-dispersive X-ray spectroscopy (EDX) elemental mapping and analysis plots corresponding to the assembled $WSe_2$ and $MoS_2$, respectively.

Modulation of the electric field is crucial in the device operation mechanism, thus it is essential to ensure the formation of depletion region and built-in electric field at interface. The work functions of $WSe_2$ and $MoS_2$ can be measured by Kelvin probe force microscopy (KPFM) to be 4.81 and 4.68 eV, respectively, with a surface potential difference (SPD) of 135 mV between both, as shown in Fig. 2d and Supplementary Fig. 3. The calculation for KPFM measurement is presented in Methods. With the lower work function of $MoS_2$, the electrons shift from $MoS_2$ to $WSe_2$ after contact until reaching an equilibrium state, forming a built-in electric field with direction pointing from $MoS_2$ to $WSe_2$, According to the above analysis and previous report[24,25], $WSe_2$/$MoS_2$ heterojunction presents a typical type-II energy band arrangement as shown in Fig. 2e[26]. The $I_{GS}$-$V_{GS}$ curve exhibits the diode characteristic with a current rectification ratio as high as $10^4$ (Fig. 2f) and an ideal factor of unity (Supplementary Fig. 4), further demonstrating the formation of the depletion region in the high quality of the heterojunction.

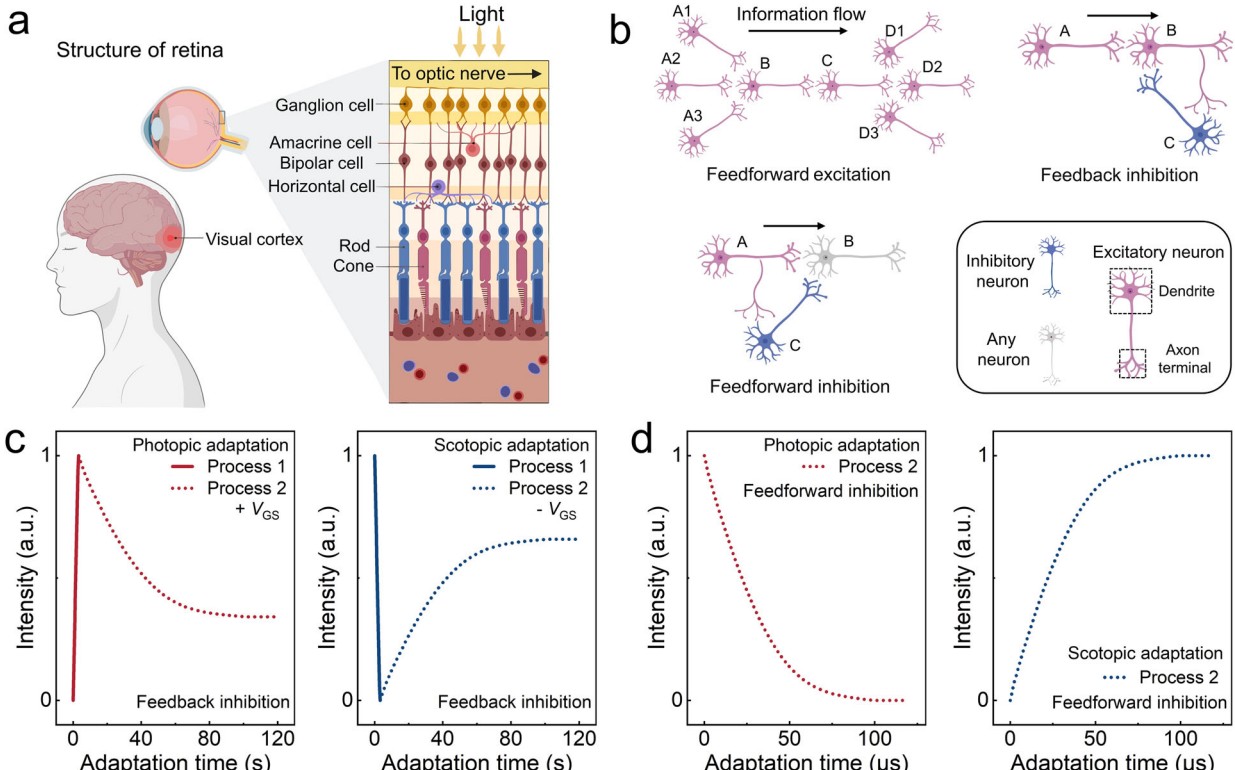

**Fig. 1 | Classical core circuit motifs in retina. a** Schematic of the retina. **b** The three classical circuit motifs. In a single neuron, the information is transmitted from the dendrite to the axon terminal, and the arrows represent the direction of information between neurons. Feedforward excitation: information is transmitted from neuron A to D, where B and C exhibit convergent and divergent properties, respectively. Feedback inhibition: neurons C and B form a circuit, and the output of C is modulated by B. Feedforward inhibition: neuron B receives both excitatory and inhibitory information. **c**, **d** Photopic and scotopic adaptation of bionic devices based on feedback inhibition (**c**) and feedforward inhibition (**d**), respectively. Figure 1a, b Created with BioRender.com released under a Creative Commons Attribution-NonCommercial-NoDerivs 4.0 International license.

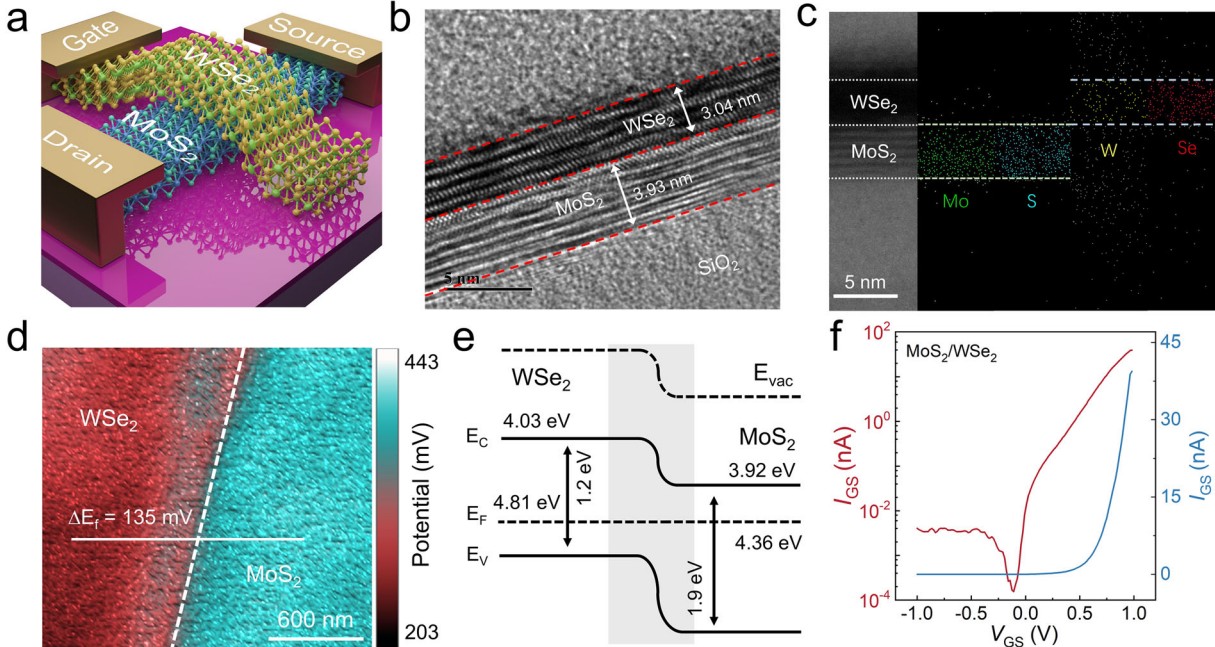

**Fig. 2 | Characterization of bionic visual device based on MoS₂/WSe₂ vdW heterostructure. a** Schematic diagram of the device. **b** Cross-sectional high-resolution transmission electron microscope image of the constituent layers. **c** Scanning transmission electron microscopy (STEM) image and corresponding energy-dispersive X-ray spectroscopy elemental (EDS) mapping image of the fabricated device. **d** In situ surface potential difference (SPD) image of the MoS₂/WSe₂ interface. **e** Energy band diagram of MoS₂/WSe₂ at equilibrium state. The $E_F$, $E_C$, and $E_V$ are the Fermi level, conduction band, and valence band, respectively. **f** $I$-$V$ characteristics of MoS₂/WSe₂ vdW heterostructure.

## Gate-tuned avalanche properties and operation mechanism

Acting as a JFET, our device exhibits a well-switching behavior as demonstrated by its transfer characteristics (Supplementary Fig. 5). Interestingly, by further increasing the drain voltage ($V_{DS}$), an avalanche phenomenon is observed as shown in Fig. 3a. Under a fixed gate voltage and with an increase in $V_{DS}$, the drain current ($I_{DS}$) deviates from linear to reach saturation due to the pinch-off effect, then increases rapidly as the carrier avalanche multiplication when $V_{DS}$

exceeds the electric breakdown voltage ($V_{EB}$). Notably, the gate voltage ($V_{GS}$) exhibits significant modulation on $V_{EB}$ and multiplication factor as shown in Fig. 3b. The definition of the $V_{EB}$ is shown in Supplementary Fig. 6 and the multiplication factor is defined as $M = \frac{I}{I_s}$, where $I$ and $I_s$ are the avalanche and saturation current, respectively. As $V_{GS}$ decreases, the $V_{EB}$ is lowered while the multiplication factor is improved significantly, reaching 5.48 V and $5.29 \times 10^5$, respectively, at $V_{GS} = -3$ V. Figure 3c summarizes $V_{EB}$ and multiplication factor of the reported 2D

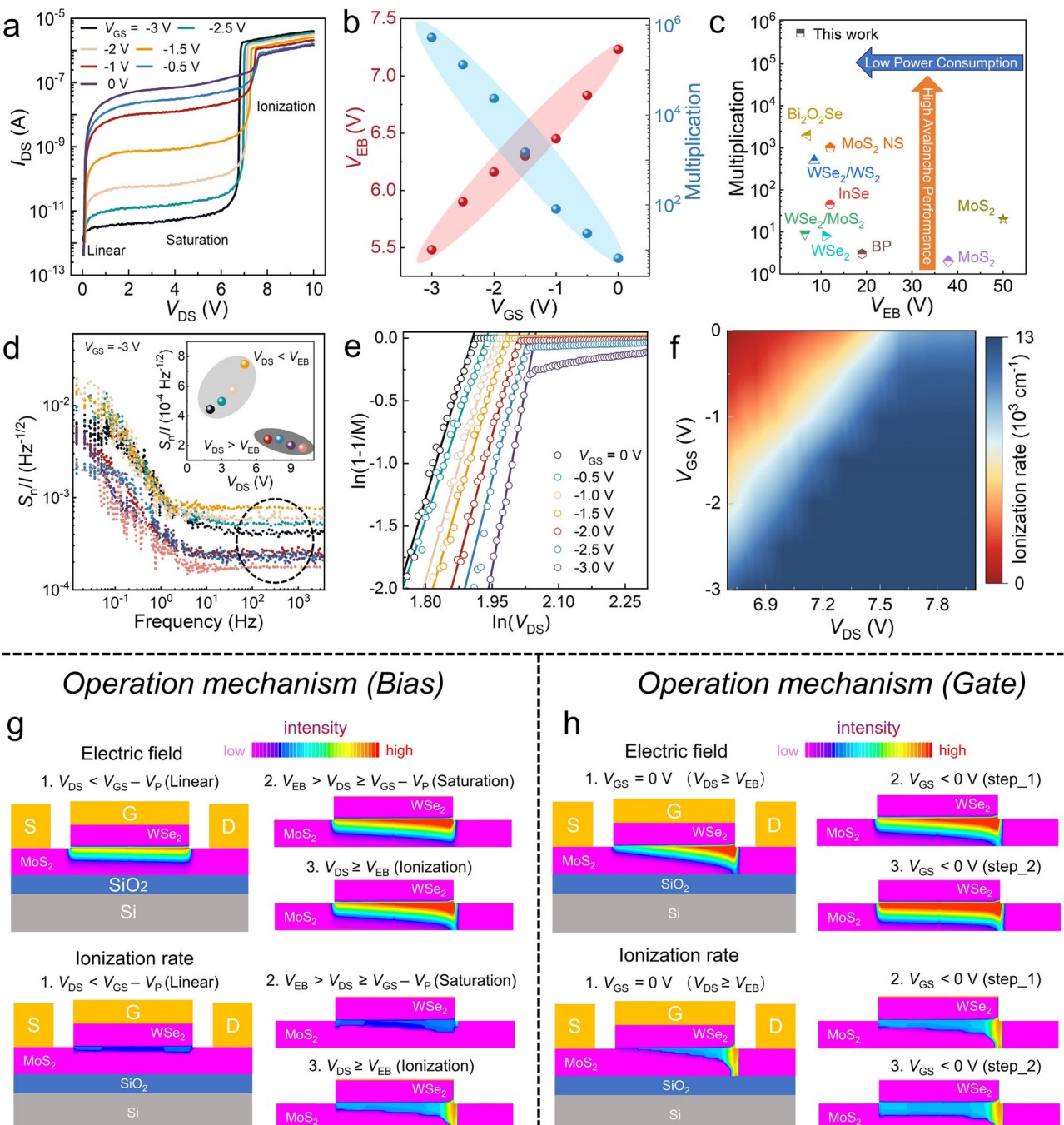

**Fig. 3 | Avalanche properties and operation mechanism. a** Output characteristics at different gate voltage ($V_{GS}$) in the logarithmic scale of the y-axis. **b** The electrical breakdown voltage ($V_{EB}$) and Mutiplication factor as a function of $V_{GS}$. **c** $V_{EB}$ and Multiplication factor of our device and previously reported 2D devices. **d** $S_n/I$ as a function of frequency for different bias voltages at $V_{GS} = -3$ V. $S_n$ and $I$ are the noise

spectral density and the corresponding current, respectively. **e** Logarithmic plot of 1 - 1/M versus $V_{DS}$ in avalanche multiplication at different $V_{GS}$. **f** Ionization rate mapping at different bias and gate voltage. **g, h** Technology computer-aided design (TCAD) simulated the evolution of the channel electric field (top panel) and ionization rate (bottom panel) by varying (**g**) $V_{DS}$ and (**h**) $V_{GS}$.

avalanche transistors, showing the superior avalanche performance and lower power consumption of our device with highest multiplication factor and low breakdown voltage[27–35].

Detection sensitivity is a key parameter to evaluate the avalanche properties and is inversely proportional to noise. Figure 3d and inset shows the noise density spectrum to current ($S_n/I$) at different $V_{DS}$, transforming from $1/f$ noise to stable white noise as increasing frequency. Interestingly, when an avalanche occurs at $V_{DS} > V_{EB}$, the sensitivity is improved with decreased $S_n/I$ from $2.41 \times 10^{-4}$ to $1.83 \times 10^{-4}$ Hz$^{-1/2}$, which is consistent with the Geiger mode characteristics of conventional avalanche transistors and implies that the devices have the potential for weak light detection. $S_n$ is acquired from the Fourier transformation of dark current traces, as shown in Supplementary Fig. 7. The ionization index (n) can be calculated from the formula $1 - \frac{1}{M} = (\frac{V_{DS}}{V_{EB}})^n$, which is related to the ionization rate at the drain side. By calculating the slope of the linear region in Fig. 3e, n remains nearly constant with a value of -12.2 at different $V_{GS}$, as demonstrated in Supplementary Fig. 8. Figure 3f shows the ionization rate mapping as a function of $V_{DS}$ and $V_{GS}$, which can be calculated by $\alpha(V_{DS}) = \frac{1}{n} * \frac{dn_e}{dx} = \frac{1}{L} * (1 - \frac{1}{M})$, where $n_e$ is the electron density and $L$ is the channel length. As $V_{GS}$ decreases, the breakdown voltage is reduced, and the ionization rate increases by regulating the electric field at the MoS$_2$/WSe$_2$ junction. Due to space charge limitation[36], the ionization rate gradually saturates to $1.2 \times 10^5$ cm$^{-1}$ with increasing $V_{DS}$.

A strong electric field is necessary for the avalanche effect whose strength can be represented by the impact ionization rate. Therefore, the avalanche operation mechanism can be analyzed by technology computer-aided design (TCAD) simulation on the intensity and distribution of electric field and ionization rate in the MoS$_2$ channel. The applied bias causes a voltage gradient across the channel, with the potential increasing from the source to the drain side. In this work, due to the voltage gradient within the channel, the higher potential at the drain side induces a thicker depletion region, causing the channel to form an unaligned depletion region vertically. As shown in Fig. 3g, with increasing $V_{DS}$, the electric field gradually increases to reach a pinch-off point that then moves towards the source end, which corresponds to the three working regions (linear, saturation, and ionization) in Fig. 3a, respectively. When the pinch-off point moves, the intensity and distribution of the ionization rate increase rapidly, which means that more electrons with high kinetic energy impact ionization to induce an avalanche effect. Here, the regulation of the avalanche effect by the bias primarily involves two factors, one is the increasing carrier drift velocity, and another is a thicker depletion layer at the drain side through the voltage gradient. Figure 3h illustrates $V_{GS}$ regulation on avalanche effect at ionization region. The distribution of both electric field and ionization rate is positively correlated with $|V_{GS}|$, indicating that increasing $|V_{GS}|$ can strengthen the avalanche effect. Notably, at $V_{DS} = 0$ V, as the $-V_{GS}$ increases, the depletion layer thickness increases in an aligned manner. Consequently, under the same $V_{DS}$, a larger $-V_{GS}$ results in a quicker transition to the avalanche state and forms a larger pinch-off region. However, due to the avalanche mainly occurring near the pinch-off area causing a higher ionization rate, the larger $-V_{GS}$ does not further increase the ionization rate at the drain side, which corresponds to the constant ionization index as discussed above. These TCAD simulation results are very consistent with the observed bias- and gate-tuned avalanche effect in our device.

**Light intensity-dependent avalanche and bionic neural network**
Now, we turn to how our device can behave with a bionic visual function. As shown in Fig. 4a, the output current and avalanche effect can also be largely tuned by the incident light stimulus. With increasing light illumination, the photocurrent increases as a positive photoconductivity (PPC) in the linear and saturation regions, however, the PPC gradually transitions to negative photoconductivity (NPC) in the ionization region. Figure 4b shows the photocurrent and avalanche

gain as a function of light intensity in the ionization region ($V_{DS} = 7.5$ V). The photocurrent first increases to 5.1 μA and then decreases to −2.2 μA with increasing light power, which behaves like the spontaneous visual adaptation preventing output of overstimulation information. The avalanche gain is defined as $Ava\_Gain = \frac{I_{light} - I_{dark}}{I_{light0} - I_{dark0}}$, where $I_{light}$ is photocurrent and $I_{dark}$ is dark current, $I_{light0}$ and $I_{dark0}$ are the photocurrent and dark current at $V_{DS} = V_{EB}$, respectively. As light intensity increases, the avalanche gain decreases from $1.5 \times 10^4$ to −8, indicating that the dominant photo-sensing mechanism shifts from avalanche to photoconductivity effect. Figure 4c exhibits the sensor responsivity as a function of light intensity in both ionization and saturation regions. Significantly, the responsivity in the ionization region experiences great changes in both magnitude and sign, ranging from $7.6 \times 10^4$ to $-1 \times 10^3$ A/W, while that in the saturation region varies slightly from 158 to 5 A/W. The sensitivity evolution in the ionization region is similar to that in the retina, which demonstrates the reliability of the device model[37]. Notably, the drain current is higher than the leakage current by more than $10^3$, confirming the validity of the avalanche effect and the high reliability of the device, as shown in Supplementary Fig. 9.

For human retina, by changing the environment from dim to bright, the photoreceptors including rod cells (high sensitivity) and cone cells (low sensitivity) will dominate the perceptual function alternately. The transition between avalanche and photoconductivity effect in our device is like the switchover between rod and cone cells in the retina. In this way, the avalanche tuning with varying light illumination conditions has endowed the JFET with a visual behavior. The retina sensitivity gradually changes over time for a long visual adaption process due to that the switchover of the photoreceptor cells is controlled by feedback inhibition and regeneration/bleaching photopigment[3,4]. To be in contrast, the sensitivity evolution of our device in the avalanche ionization region is light-adaptive and real-time, ensuring the immediate perception of the environment change and avoiding the potential harms caused by the long scotopic and photopic adaptation process of the human retina. It is noted that the switchover of photo-sensing mechanism is accompanied by sign reversal and magnitude changes with over five orders in both avalanche gain and responsivity, the large difference in sensitivity at weak and strong light stimuli can benefit the image contrast enhancement, which is superior to the retina and reported bionic device with visual adaption.

By comparing the sensitivity and avalanche gain with reported avalanche detectors in Fig. 4d, our device demonstrates superior avalanche photodetection characteristics with an avalanche gain of $1.5 \times 10^4$ and responsivity up to $7.6 \times 10^4$ A/W, exhibiting great potential in weak-light detection and clear visualization at dim environment acting as bionic visual sensor[27,29,31,34,35,38,39]. As shown in Fig. 4e, the ionization index increases first and then decreases with increasing light intensity, whose trend is consistent with the photocurrent evolution as observed in Fig. 4b, indicating an efficient modulation of light on the ionization rate at the drain side[40]. Figure 4f shows the ionization rate mapping, which again verifies the operation mechanism shifting from avalanche to photoconductivity. The external quantum efficiency (EQE) and $V_{EB}$ as a function of light illumination are discussed in Supplementary Figs. 10–12.

To further explain the light-tuning avalanche effect, the TCAD simulated electric field and ionization rate under different illumination conditions are shown in Fig. 4g. The built-in electric field is inversely proportional to the light power and nearly disappears under strong light, which is due to the reversed photo-generated voltage at MoS$_2$/WSe$_2$ junction to counteract the built-in electric field. On further analysis, the ionization region area first increases and then decreases with increasing light, depending on the inhibition degree of the photo-generated voltage on the avalanche gain in the MoS$_2$ channel. In weak light condition, the electric field is slightly influenced, and the avalanche gain of the photo-generated carriers dominates. Under strong

## Light intensity dependent avalanche

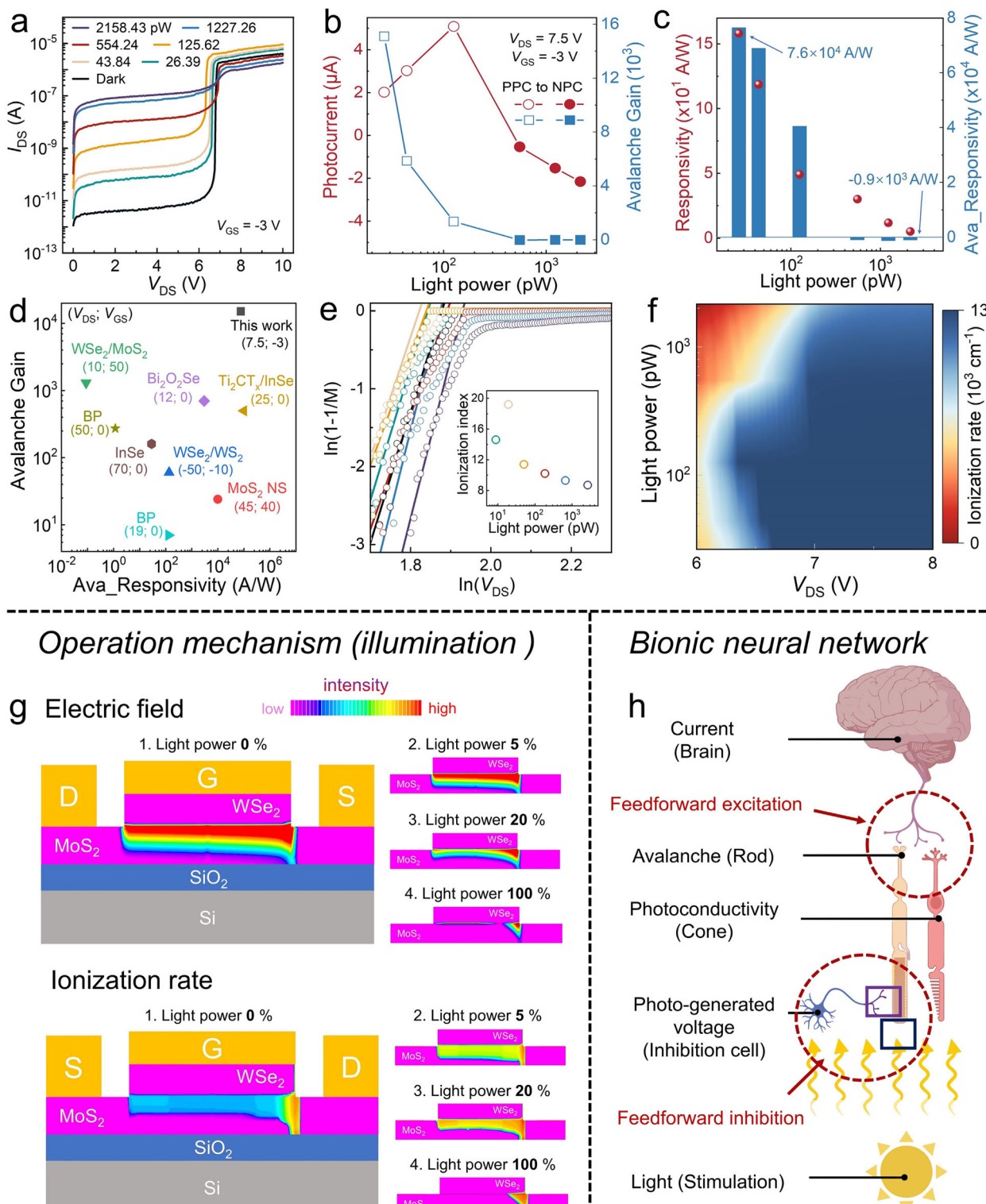

**Fig. 4 | Light intensity-dependent avalanche and operation mechanism.**
**a** Output characteristics of the device under various light power with $V_{GS} = -3$ V in the logarithmic scale of the y-axis. **b** Photocurrent and avalanche gain extracted from the output characteristics as a function of light power at $V_{DS} = 7.5$ V. The hollow and solid patterns represent the device in positive and negative photoconductivity, respectively. **c** Responsivity with $V_{DS} = V_{EB}$ and avalanche responsivity with $V_{DS} = 7.5$ V as a function of light power. **d** Comparison of avalanche gain and responsivity between our device and previously reported avalanche photodetector. **e** Logarithmic plot of 1 - 1/M versus $V_{DS}$ in avalanche multiplication under different light powers. The inset is the ionization index as a function of light power. **f** Ionization rate mapping at different bias and power. **g** TCAD simulations on the channel electric field (top panel) and ionization rate (bottom panel) at different light illumination conditions. **h** Schematic diagram of bionic neural network. Red circles represent the circuit motifs. Figure 4h Created with BioRender.com released under a Creative Commons Attribution-NonCommercial-NoDerivs 4.0 International license.

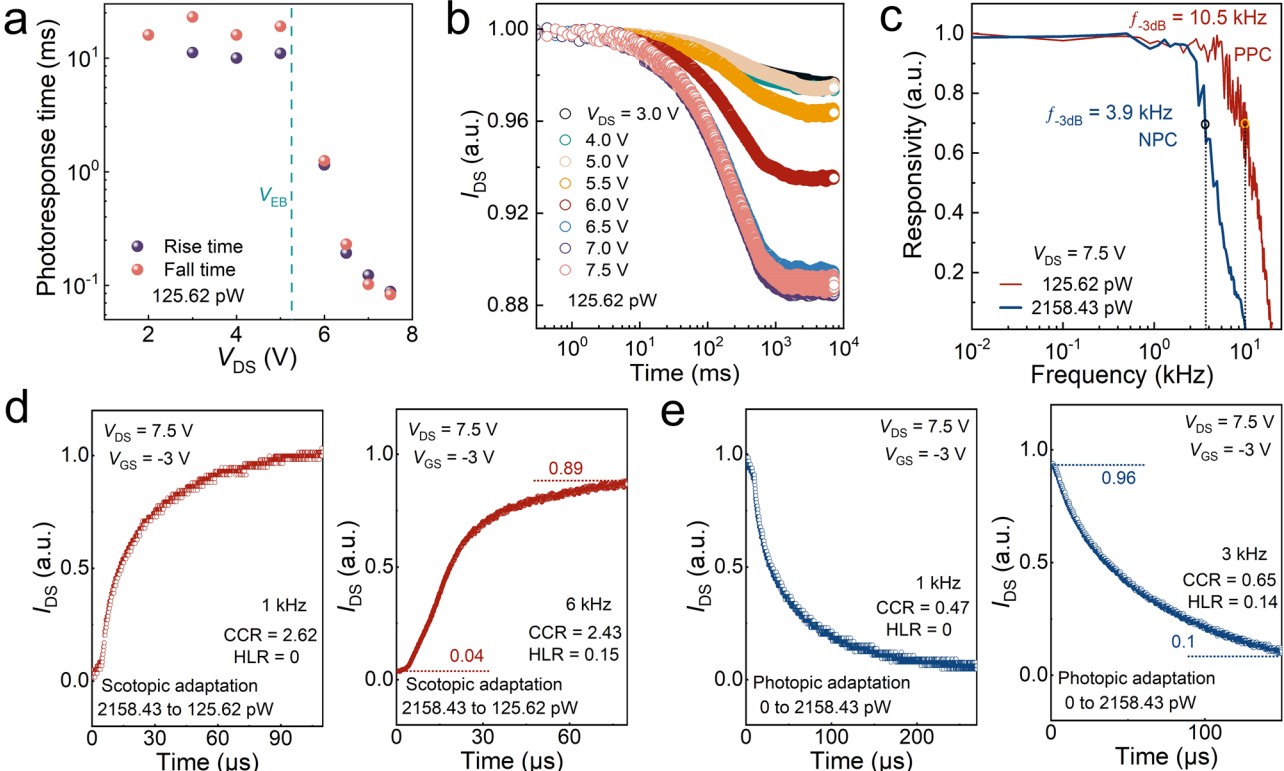

**Fig. 5 | Avalanche photoresponse properties. a** Fall and rise response time with $V_{DS}$ under light power of 125.62 pW. **b** Normalized $I_{DS}$ as a function of time with different fixed $V_{DS}$ under continuous 125.62 pW light illumination. **c** Normalized NPC and PPC response bandwidths at $V_{DS}$ = 7.5 V, where the vertical dashed lines represent the −3 dB bandwidth. **d**, **e** Normalized real-time dependent current with different bandwidths under $V_{DS}$ = 7.5 V in background illumination of (**d**) 125.62 and (**e**) 2158.43 pW, respectively. The initial conditions for scotopic and photopic adaptation are set to strong light power (2158.43 pW) and darkness, respectively.

light illumination, the electric field is significantly weakened and accompanied by a gradual disappearance of the depletion region, which can subsequently inhibit the avalanche effect and reduce the photoresponse sensitivity. The TCAD simulations on electric field and ionization rate can match well with the experimental results, manifesting that the light-tuning avalanche effect can emulate the visual adaptation function.

Distinct from previous 2D bionic devices, our device can optimize visual adaptation beyond the retina by introducing an efficient bionic neural network, as shown in Fig. 4h. In this network, the avalanche and photoconductivity effect can emulate the function of rod and cone cells, respectively, because the photosensitivity by avalanche effect is four orders higher than that by photoconductivity effect. The photo-generated voltage is opposite in direction to the built-in electric field at the $MoS_2$/$WSe_2$ junction, which can be seen as an inhibition cell modulating the avalanche effect. Light (Stimulation), "Photo-generated voltage" (Inhibition cell), and "Avalanche" (Rod) can form a feedforward inhibitory circuit where the "Avalanche" receives both stimulus and inhibitory information to avoid it outputting over-current under photopic adaptation conditions. At strong light illumination conditions, the avalanche effect is inhibited, turning the photo-sensing mechanism to "Photoconductivity" (Cone). On the contrary, the mechanism switches from "Photoconductivity" to "Avalanche" by changing light stimulus to weak condition, corresponding to the scotopic adaptation process. For both photopic and scotopic adaptation, the switchover between "Photoconductivity" and "Avalanche" is much faster than the switching process between cone and rod through chemical reactions in the retina. Notably, the feedforward excitation circuit formed by the "Output current", "Avalanche" and "Photoconductivity" exhibits multiplexed modulation characteristics and effectively improves the signal-to-noise ratio. Thus, our device offers

great advantages in visual adaptation compared to human retina and previously reported bionic sensors by introducing a feedforward circuit as fast-switching mechanism.

## High-frequency visual adaptation

The dynamic photoresponse at different $V_{DS}$ and modulation frequency of incident light has been measured as shown in Supplementary Fig. 13, and the rise/fall times are extracted and plotted in Fig. 5a. Before the breakdown, the device works in photoconductive mode and exhibits relatively slow response speed of ~15 ms due to the long-lived trap states. When $V_{DS}$ > $V_{EB}$, the device switches to avalanche mode with a much faster speed of 88 μs, which can be attributed to the large number of avalanche-generated carriers filling the trap states in the $MoS_2$ channel. To verify that, we measured the time-varying $I_{DS}$ at different fixed $V_{DS}$ under continuous light illumination, as shown in Fig. 5b. As time prolongs, the current drops significantly in the range of $10^1$–$10^2$ ms, which corresponds to the release of trapped holes from shallow level defects in $MoS_2$ channel[41]. When $V_{DS}$ exceeds $V_{EB}$, the current dropping effect is more noticeable due to more released holes in the avalanche region.

Figure 5c demonstrates the response bandwidth in weak and strong light conditions, which represent the timescale of visual perception in bionic vision devices. The dynamic response of the human retina is 2 ms, corresponding to a bandwidth of 500 Hz. The −3 dB bandwidth of our device in the avalanche region can reach 10.5 and 3.7 kHz under weak and strong light, respectively, which are much superior to the retina and reported 2D JFETs detectors[42–44]. The bandwidth under weak light is twice larger than that under strong light, which is related to the inhibition of the avalanche effect by the photo-generated voltage at strong light illumination. At $V_{DS}$ of 3 V, the device is operated in photoconductive mode, showing a negligible light intensity

dependence of dynamic response (Supplementary Figs. 14, 15). Notably, the device exhibits PPC and NPC effect under weak and strong light illuminations, respectively, which is corresponding to the current excitation and inhibition, respectively, enabling the visual adaptation function.

To further verify the advantage of the bionic device in visual adaptation, Fig. 5d, e shows the normalized real-time dependent current of the device under the simulated high-frequency scotopic and photopic adaptation conditions. For scotopic adaptation (Fig. 5d), the environmental illumination is switched from bright (2158.43 pW) to dim (125.62 pW) conditions, subsequently the current increases over time because the avalanche effect takes over the photo-sensing mechanism for higher sensitivity, which is analogous to the role of rod cells for higher visual sensitivity of photoreceptor over time under the dim-light condition. To emulate photopic adaptation (Fig. 5e), the dark background is suddenly turned to be bright, the current decreases quickly over time under continuous strong light irradiation due to the inhibition of "Avalanche" by the "Photo-generated voltage" as discussed above, which is like the inhibitory effect on rod cells by photopigment bleaching in the retina during the photopic adaptation process.

Unlike the long adaptation process of the retina, the current of our bionic device can reach saturation states within 108 and 268 µs for scotopic and photopic adaptation conditions, respectively, at a light frequency of 500 Hz, indicating the ultra-fast visual adaptation, which is of great importance for rapid response and emergency operation of machine vision systems upon the complex environment. Due to the fast visual adaptation process, our device can work to emulate both scotopic and photopic adaptation at 6 and 3 kHz, respectively, without much signal loss. High-frequency signal loss ratio (HLR) is a crucial parameter in the vision system, which is defined as $\text{HLR} = \left(\frac{I_{\text{ini}} - I_{\text{ini0}}}{I_{\text{ini0}}}\right) + \left(\frac{I_{\text{fin0}} - I_{\text{fin}}}{I_{\text{fin0}}}\right)$, where $I_{\text{ini}}$ and $I_{\text{fin}}$ represent the initial and final currents at high frequency (6 or 3 kHz), respectively; $I_{\text{ini0}}$ and $I_{\text{fin0}}$ represent the initial and final currents at low frequency of 500 Hz, respectively. The HLR values for scotopic and photopic adaptation are calculated to be only 15 and 14%, respectively, again verifying the more efficient image information processing capability at high-frequency conditions.

The current change ratio (CCR), defined as $\text{CCR} = \frac{I_{\text{Fin}}}{I_{\text{Ini}}}$, where $I_{\text{Ini}}$ and $I_{\text{Fin}}$ represent the current at the initial and final state, is also proposed to quantitatively analyze the current excitation and inhibition effect. At 500 Hz, the CCR values are 2.62 and 0.47, which are larger and less than 1 for scotopic and photopic adaptation, respectively, indicating the current excitation and inhibition effect, respectively. At higher frequency (6 or 3 kHz), the slight variation in CCR values suggests that the visual adaptation function of our device is also reliable at high frequencies of environmental light. Supplementary Table 1 summarizes the reported negative photoconductive devices with potential applications in visual adaptation to highlight the advantages of our device. Overall, benefiting from the avalanche tuning mechanism as feedforward inhibition circuit, the bionic device exhibits fast and high-frequency visual adaptation behavior, which is much superior to the human retina and widely reported bionic sensors relying on feedback inhibition circuit. More details about the frequency, light intensity, and durability-dependent photoresponse measurements are shown in Supplementary Figs. 16–21.

**Machine vision with ultra-fast and accurate image recognition**
Deep learning with convolutional neural network (CNN) plays a crucial role in the image recognition function of machine vision, but the image brightness has a significant impact on accuracy. In Fig. 6a, to assess the image recognition performance of a typical three-layered CNN under varying brightness conditions, we utilized 60,000 MNIST dataset images with different brightness levels as the training set to CNN

30 times. Notably, during the training process, we input image brightness as an explicit additional parameter into the network. The detailed process of CNN deep learning is presented in Methods. As shown in Fig. 6b, the network exhibits excellent robustness and maintains an accuracy of 98.3% during the almost whole brightness-decreasing process. However, the accuracy declines significantly with increasing brightness, which is attributed to that the neural network struggles to accurately capture crucial features of over-exposed images. In quantifying the influence of brightness augmentation on image recognition, the confusion matrices in Fig. 6c, d present the results of 10,000 image recognition trials under standard and 20% increased brightness conditions, respectively. It's concerning that under the +20% brightness condition, the accuracy is only 83%, indicating that CNN are unable to improve the accuracy of classification features even though they have obtained brightness parameters on a fixed dataset. More confusion matrices at various brightness are presented in Supplementary Fig. 22.

To broaden the brightness scope of image perception and improve the recognition accuracy under bright conditions, we have employed a strategy that combines a convolutional neural network with our bionic $MoS_2/WSe_2$ transistors to construct an adaptative machine vision. This system possesses precise image recognition capabilities based on CNN and achieves ultra-fast visual adaptation via a bionic transistor. In Fig. 6e, f, we select the number "7" as the test image feature, with the illumination provided by the laser and the brightness data obtained by current mapping subsequently imported to CNN for processing. Under scotopic adaptation, the accuracy soars to 98.3% in a mere 9.5 µs, and it rapidly climbs to 98.2% in just 174 µs during photopic adaptation. This validates the adaptative machine vision with rapid adaptation and precise image recognition capabilities in different brightness environments. The inset depicts the contrast changes of MNIST images during the adaptation process, emphasizing the required image contrast for the adaptative machine vision system to efficiently capture and analyze image features. In a nutshell, the adaptative machine vision, with its microsecond-level adaptation time, significantly enhances image recognition accuracy with ultra-fast speed under varying brightness and environmental conditions. It holds vast potential across critical application scenarios such as facial recognition and autonomous driving, allowing for swift adjustments to the desired brightness and atmospheric conditions, thereby improving the efficiency of real-time image processing. Furthermore, there is the prospect of expanding the brightness scope of image perception, simplifying the complexity of hardware and algorithms, thereby enhancing the image processing capabilities of sensor terminals and propelling further advancements in machine vision technology.

## Discussion
We have fabricated a bionic visual transistor that can be used to emulate the visual adaptation beyond the retina. Avalanche effect occurs by the impact ionization in the $MoS_2$ channel when the bias exceeds breakdown voltage ($V_{\text{EB}}$ -5.48 V). Through TCAD simulation, the electric field and ionization rate can be tuned by both $V_{\text{DS}}$ and $V_{\text{GS}}$, enabling the tunability of avalanche performance. Under light illumination, the dominant mechanism responsible for photocurrent generation can be controlled by both $V_{\text{DS}}$ and light intensity. By varying $V_{\text{DS}}$ (larger or smaller than $V_{\text{EB}}$) and light intensity (from weak to strong), the photo-sensing mechanism of the device is switched from avalanche to photoconductive effect, resulting in a change of sensitivity from high to low value, whose function is like the rod and cone cells of the photoreceptor in the retina. The feedback inhibition causes a long visual adaption process of retina, for further optimization, we introduce a feedforward inhibition in our bionic device-based neural network by using the avalanche tuning operation, realizing a light-adaptive and real-time vision behavior. This can enable the visual device to timely perceive the rapid change of brightness, avoiding the

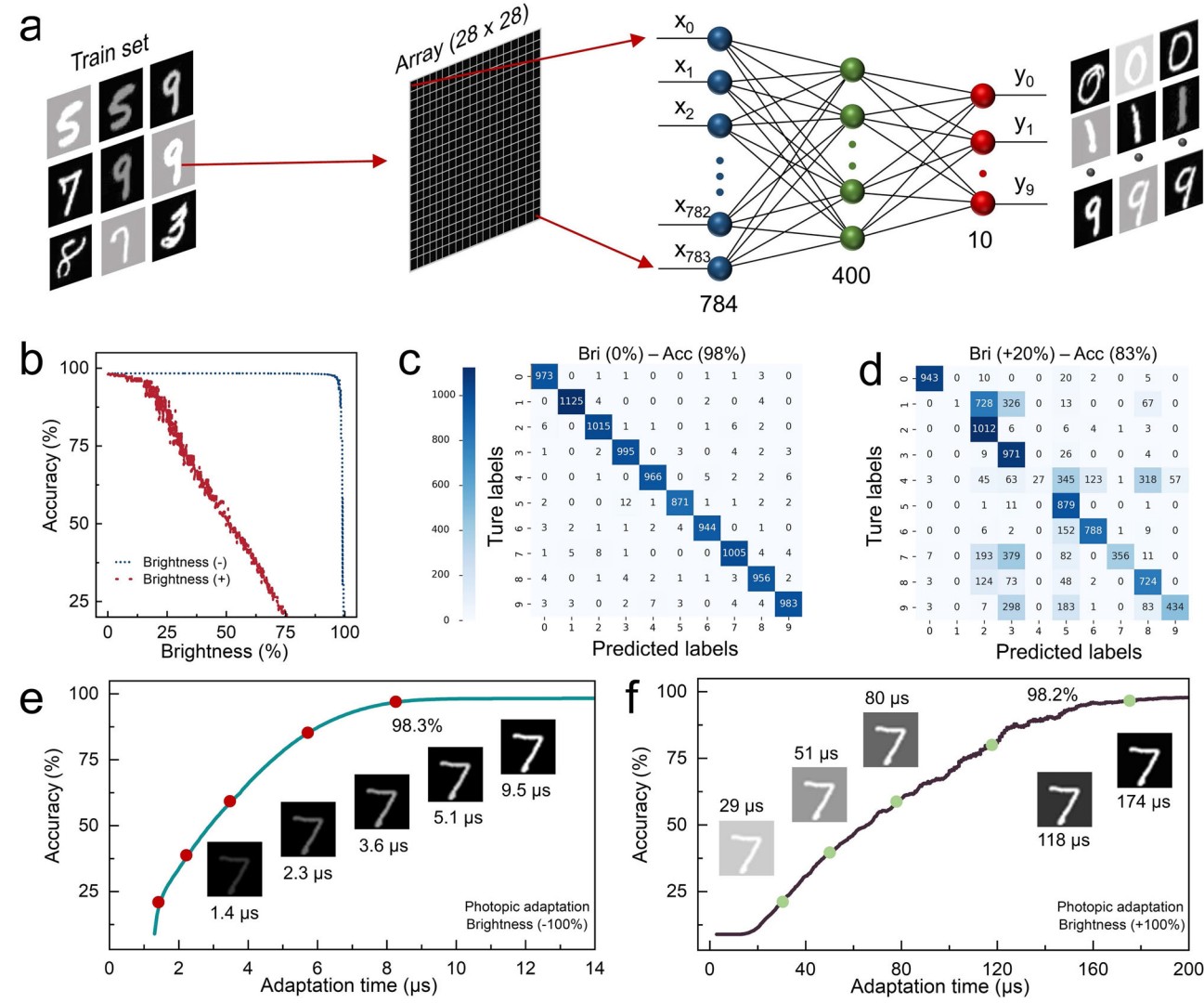

**Fig. 6 | Adaptative machine vision. a** Illustration of the machine vision based on a convolutional neural network for image recognition. **b** The image recognition rate as a function of brightness conditions. The minus and plus signs represent the brightness condition decreasing and increasing compared to the standard brightness, respectively. **c**, **d** Confusion matrix with 10,000 test results in (**c**) standard and (**d**) +20% brightness conditions. **e**, **f** Recognition rate of adaptative machine vision as a function of time for (**e**) scotopic and (**f**) photopic adaptation. The inset shows the results of the visual adaptation simulation for MNIST image "7".

occurrence of potential harms such as car accidents. The TCAD simulation on the distribution of electric field and impact ionization rate has also been performed to further verify the bias- and light-tuning avalanche effect. In addition to the high avalanche gain of $1.5 \times 10^4$ and responsivity of $7.6 \times 10^4$ A/W in the avalanche ionization region, our device also exhibits a large −3 dB bandwidth up to 10.5 kHz at weak light (125.62 pW). The bionic device achieves ultra-fast and high-frequency visual adaptation behavior at 6 and 3 kHz for simulated scotopic and photopic adaptation conditions. Importantly, through the combination of convolutional neural networks with bionic avalanche transistors, an adaptative machine vision system has been achieved. This system demonstrates exceptional microsecond-level rapid adaptation, enabling image recognition with over 98% precision in both dim and bright lighting conditions. Thus, the avalanche tuning based bio-inspired visual device can avoid long time visual adaptation process by introducing a more predictive and faster feedforward inhibition circuit, that holds great promise for widespread applications in the field of machine vision, bringing forth ideas and designs for bio-inspired visual systems while avoiding excessive reliance on complex circuits and algorithms.

## Methods

### Device fabrication

Few-layer $MoS_2$ and $WSe_2$ were mechanically exfoliated from the crystal (Shanghai OnWay Technology Co., Ltd). The 3.93 nm $MoS_2$ and 3.04 nm $WSe_2$ were successively stacked on 300 nm $SiO_2$/Si substrate using the PVA (polyvinyl alcohol, MACKLIN Co., Ltd, Shanghai)/PDMS (poly-dimethylsiloxane) assisted dry transfer technique. The substrate with the photoresist (An ARP-5350 positive photoresist from Taizhou SUNANO New Energy Co., Ltd.) was spin-coated onto the substrate at 3000 rpm for 60 s and then baked on a hot plate for 4 min at 100 °C. The Cr/Au (5/50 nm) electrodes were fabricated using an Ultraviolet Maskless Lithography machine (TuoTuo Technology, UV Litho-ACA) and an electron beam evaporation technique. Finally, the device was annealed for 2 h at 150 °C under $N_2$ atmosphere to improve the interfacial contact.

### Device characterization

The potential difference of the heterojunction interface was measured by a Scanning Probe Microscope (SPM) with the functional modules of KPFM (Oxford Cypher S AFM. Co., Ltd.), which can be calculated by

following equation: $e*SPD = e*\phi_{tip} - e*\phi_{sample}$, where $e$ is elementary charge, $e\phi_{tip}$ and $e\phi_{sample}$ are the work functions of AFM tip and sample, respectively. The STEM and energy-dispersive X-ray spectroscopy (EDX) elemental mapping of $MoS_2$ and $WSe_2$ were measured using STEM (Thermo Fisher Talos F200X). The electrical characterizations of the device were measured via a four-probe station (PSAICPB6A, PRECISION SYSTEM INDUSTRIAL Co., Ltd.) equipped with a KEITHLEY 2636B and 2611B semiconductor source meter. The noise spectral density was measured via KEITHLEY 2636B and oscilloscope then calculated by the Fourier transformation. The photoresponse was tested using a 635 nm optical fiber laser and the spot diameter of all the excitation lasers was ~3 mm. The response time was extracted via an electric shutter system and oscilloscope. The rise and fall times are defined as the time it takes for the current to rise from 10 to 90% and the time it takes for the current to fall from 90 to 10%, respectively.

## Deep learning

In the deep learning process of convolutional neural networks, the input encompasses grayscale information from 784 pixels of an image. This information undergoes processing through fully connected layers, enabling the synthesis and extraction of individual pixel features. Eventually, the input pixel information is mapped into the network, forming a hidden layer. Through this deep learning process, the network can accurately classify training images and learn the patterns and features within them. The role of the fully connected layer is to interactively process various parts of the image, forming a holistic understanding of the image. This comprehensive processing equips the network with the ability to recognize complex patterns and abstract features, enabling it to learn both surface-level image characteristics and deeper layers of image structure during the training process.

## Data availability

The data supporting the findings of this study are available within the article and its supplementary files. Any additional requests for information can be directed to, and will be fulfilled by, the corresponding authors. Source data are provided with this paper.

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

## Acknowledgements

We acknowledge financial support from the Guangdong Basic and Applied Basic Research Foundation (No. 2024A1515030107), the National Natural Science Foundation of China (Nos.11904108 and 62004071), the China Postdoctoral Science Foundation (No.2020M672680), the "Pearl River Talent Recruitment Program" (No.2019ZT08X639), and the Scientific Research Start-up Foundation for PhD of Chaohu University (No. KYQD-2023012). We acknowledge BioRender.com for helping us to create Figs. 1a, 1b, 4h.

## Author contributions

L.L. and S.L. contributed equally to this work. L.L. and N.H. conceived the idea and supervised the work. W.W., Y.S. and M.Y. designed the experiments. J.Z., Q.D., T.Z., J.L. and W.G. helped with mechanism analysis and discussion. H.W. and Y.P. analyzed the physical model. X.L. performed device fabrication and characterization. Y.Y. supports the characterization of materials. L.L., J.L. and N.H. co-wrote the manuscript and all authors contributed to the revision of the manuscript.

## Competing interests

The authors declare no competing interests.
