## [Peer Review File · Nature Communications]

Adaptative machine vision with microsecond-level accurate perception beyond human retinaReviewers' comments:

Reviewer #1 (Remarks to the Author):

In this manuscript, the authors propose a junction field effect transistor based on MoS₂/WSe₂ vdW heterostructure for adaptive light sensing. By increasing the drain voltage V_{ds} , the JFET switches from a photoconductive state to an avalanche state, so that the photocurrent of the device would change from a positive photoconductivity (PPC) in the linear and saturation regions to a negative photoconductivity (NPC) in the ionization region. The authors claim that the JFET, which has a feedforward inhibition process of 140 and 427 μ s is a fast sense-computing integrated device. This paper is carefully organised with rich data results. However, due to the lack of novelty and significant insights in this paper, I do not recommend its publication in this comprehensive and impactful journal.

1. As the authors mentioned in the abstract, the key scientific issue in machine vision is the 'sluggish adaptation process making integration challenging'. However, this paper only showed a single vdW heterostructure device without any practical attempt to address the integration or system transfer challenges. The experimental design based on mechanical exfoliation and the basic findings are not new. Thus, I do not think it provides a novel suggestion to the community.
2. The fundamental physical principle comes from the negative photoconductivity (NPC). As the core performances of all are shown in Figure 5 d&e, the photocurrent increases with weakened light and decreases with stronger light. It has been widely reported that NPC is easily achieved in 2D material systems. But here the operation of the device requires applying 20 V. This is a disadvantage compared to other works presenting the NPC phenomenon. Hence, this type of device has very little potential to raise the interest of the electronics industry.
3. One of the main drawbacks of this study, which cannot be remedied in a revision, is that the image recognition simulation part of the machine vision can be completely separated from the device part. This is because the software models can be arbitrarily modelled to obtain theoretical configurations of the network, including the input image brightness conditions if the brightness conditions are affecting the network performance in some way. In other words, the machine vision computing function still relies on the software processing, and the perception accuracy is irrelevant to the device to a large extent.
4. Technical question. The device operates under a high voltage bias to reach the 'avalanche' state. Due to its JFET structure, the gate leakage current is important for reliability and power consumption. In Figure 2f, there is only a small range of gate currents. As the V_{ds} is up to 20V, the gate bias voltage also suffers more than 20V. It is necessary to present the corresponding range of gate leakage current for a further judgment of whether the avalanche effect is valid or not.

Reviewer #2 (Remarks to the Author):

This manuscript demonstrated a sense-computing integrated junction-field-effect transistor (JFET) that can be used to emulate the visual adaptation beyond retina. The photodetector based on MoS₂/WSe₂ vdW heterostructure exhibits an extraordinary avalanche performance with low breakdown voltage of \sim 10 V and high multiplication factor of 8.2×10^3 . In addition, the responsivity of the photodetector presents great changes in both magnitude and sign due to spontaneous transition of the photo-sensing mechanism between avalanche and photoconductive effect. Besides, the device shows ultra-fast adaptation process of 142 and 427 μ s under simulated scotopic and photopic adaptation conditions, respectively. Furthermore, an ultra-fast adaptive machine vision has been achieved with respect to its remarkable microsecond-level rapid adaptation capabilities and robust image recognition with over 97% precision in both dim and bright condition. These results are interesting and the paper was well organized. Therefore, it can be published after the following major revisions:

Below are the detailed comments:

1. In Supplementary Figure 2a, why is the ratio of Mo and S not close to 1:2? In addition, why are the curves in Supplementary Figure 2 and Supplementary Figure 3 not smooth?
2. From Figure 3a and Supplementary Figure 5, it can be observed that the dark current of the fabricated JFET device is relatively large (\sim 1 nA), and the I_{on}/I_{off} ratio is relatively low (\sim 103). What could be the reasons for this? Are there any methods to reduce dark current and improve the I_{on}/I_{off} ratio?

3. From the simulated results in Figure 3g, it can be observed that as VDS increases, the ionization rate in MoS2 gradually distributes away from the WeS2/MoS2 interface. What is the reason for this? In addition, why does VDS regulate the avalanche effect?
4. Regarding the simulation results in Figure 3h, the authors claim that the ionization intensity at drain-side is less modulated due to that VDS far exceeds VGS. It is recommended to provide a more detailed explanation.
5. In Figure 4a, when VDS is 20V, the dark current IDS of the device exceeds 1uA, which may result in a relatively high power consumption. How do the authors balance power consumption and photoelectric performance?
6. Can this adaptation strategy extended to dynamic motion (Nature Nanotechnology, 2023, 18, 882-888; Nature Electronics, 2023, 6, 870-878)?
7. The responsivity of the photodetector is generally significantly influenced by the applied bias. typically, a higher bias leads to a larger responsivity, but it can also introduce higher dark current. For the comparison in Figure 4d, it is suggested that the author should add bias information.
8. Is the photogenerated voltage claimed by the authors the voltage formed by the separation of photogenerated electron-hole pairs?
9. The descriptions of Figures 5d and Figure 5e in the paper seem to be written in reverse. It is recommended to check them carefully.

Response to Reviewers

We feel very grateful to the advice from reviewers on our manuscript which has been submitted to *Nature Communications* (ID: NCOMMS-24-01836-T). In response to these comments and questions, we have made a corresponding revision. We hope these are acceptable to allow the manuscript to be published.

The corresponding amendments to meet the referees' concerns have been marked in the revised manuscript. Our replies to the reviewers' suggestions and the corresponding changes are listed below.

The following are details of our response to editor and reviewers' comments:

Reviewer #1:

Comments:

In this manuscript, the authors propose a junction field effect transistor based on MoS₂/WSe₂ vdW heterostructure for adaptative light sensing. By increasing the drain voltage V_{DS} , the JFET switches from a photoconductive state to an avalanche state, so that the photocurrent of the device would change from a positive photoconductivity (PPC) in the linear and saturation regions to a negative photoconductivity (NPC) in the ionization region. The authors claim that the JFET, which has a feedforward inhibition process of 140 and 427 μ s is a fast sense-computing integrated device. This paper is carefully organised with rich data results. However, due to the lack of novelty and significant insights in this paper, I do not recommend its publication in this comprehensive and impactful journal.

Response:

We are deeply grateful for your constructive comments and suggestions, which have been instrumental in enhancing the quality of our manuscript. In response to the

concerns you raised, we have addressed the following points you concerned: (1) The significance of our work in the field of visual adaptation. (2) The superiority of our device compared to other biomimetic vision devices with NPC effects. (3) The necessity of the visual adaptation function in CNNs. (4) The range of device leakage current. We are confident that the analysis provided in our response can satisfactorily address your queries. In addition, to address your concerns of the high power consumption of our device, we have re-fabricated and optimized the device structure (see **Answer 2** for details), resulting in much improved avalanche and photo-sensing performance while reducing power consumption (the applied bias for adaptation is reduced from 20 V to 7.5 V). In this revision, we replaced all the raw data with optimized data. Overall, the quality of the revised manuscript has been improved dramatically following the constructive suggestions from the reviewers, and thanks again for the professional comments you gave.

Correction list in the revised manuscript:

1. In page 11, “**Fig. 3 | Avalanche properties and operation mechanism. a**, Output characteristics at different gate voltage (V_{GS}) in logarithmic scale of y-axis. **b**, The electrical breakdown voltage (V_{EB}) and Multiplication factor as a function of V_{GS} . **c**,

V_{EB} and Multiplication factor of our device and previously reported 2D devices. **d**, S_n/I as a function of frequency for different bias voltages at $V_{GS} = -3$ V. S_n and I are the noise spectral density and the corresponding current, respectively. **e**, Logarithmic plot of $1 - 1/M$ versus V_{DS} in avalanche multiplication at different V_{GS} . **f**, Ionization rate mapping at different bias and gate voltage.” has been revised.

2. In page 15, “**Fig. 4 | Light intensity dependent avalanche and operation mechanism.** **a**, Output characteristics of the device under various light power with $V_{GS} = -3$ V in logarithmic scale of y-axis. **b**, Photocurrent and avalanche gain extracted from the output characteristics as function of light power at $V_{DS} = 7.5$ V. The hollow and solid patterns represent the device in positive and negative photoconductivity, respectively. **c**, Responsivity with $V_{DS} = V_{EB}$ and avalanche responsivity with $V_{DS} = 7.5$ V as function of light power. **d**, Comparison of avalanche gain and responsivity between our device and previously reported avalanche photodetector. **e**, Logarithmic plot of $1 - 1/M$ versus V_{DS} in avalanche multiplication under different light powers. The inset is the ionization index as function of light power. **f**, Ionization rate mapping at different bias and power.” has been revised.

3. In page 20, “**Fig. 5. Avalanche photoresponse properties.** **a**, Fall and rise response time with V_{DS} under light power of 125.62 pW. **b**, Normalized I_{DS} as function of time with different fixed V_{DS} under continuous 125.62 pW light illumination. **c**, Normalized NPC and PPC response bandwidths at $V_{DS} = 7.5$ V, where the vertical dashed lines represent the -3 dB bandwidth. **d**, **e**, Normalized real-time dependent current with different bandwidth under $V_{DS} = 7.5$ V in background illumination of (**d**) 125.62 and (**e**) 2158.43 pW, respectively. The initial conditions for scotopic and photopic adaptation are set to strong light power (2158.43 pW) and darkness, respectively.” has been revised.

4. In page 24, “**Fig. 6. Adaptive machine vision.** **e**, **f**, Recognition rate of adaptive machine vision as a function of time for (**e**) scotopic and (**f**) photopic

adaptation. The inset shows the results of the visual adaptation simulation for MNIST image "7".” has been revised.

Question 1: As the authors mentioned in the abstract, the key scientific issue in machine vision is the ‘sluggish adaptation process making integration challenging’. However, this paper only showed a single vdW heterostructure device without any practical attempt to address the integration or system transfer challenges. The experimental design based on mechanical exfoliation and the basic findings are not new. Thus, I do not think it provides a novel suggestion to the community.

Answer 1:

We express our profound gratitude for your insightful comments. We wish to clarify that the term "integration" primarily denotes the combination of bionic vision sensor with convolutional neural networks, rather than the assembly of arrays. The object of "integration" is repeatedly mentioned in our manuscript, for example on page 3 “More importantly, an ultra-fast adaptive machine vision has been achieved by integrating convolutional neural networks with bionic avalanche transistor” and page 26 “Importantly, through the integration of convolutional neural networks with bionic avalanche transistor”. To prevent any potential misunderstandings by readers, we have changed the term "integration" to "combination."

Your remarks aptly emphasize the critical role of arrays in image capture and recognition systems, a point with which we wholeheartedly concur. Nonetheless, the large-scale arrays depend on the superior functionality and operational mechanisms of individual device. At present, the predominant challenge encountered by adaptive device is the long adaptation time and the requirement to manually adjust the gate sign according to the environment. By a rapid switch between avalanche and photoconductivity effect, we have achieved a four-order magnitude improvement in adaptation time. Notably, the transition between scotopic and photopic adaptation is autonomously triggered by light intensity, representing a significant advancement in the field of adaptive device.

Moreover, the operational mechanism we introduced possesses considerable versatility (validated through TCAD simulations), extending its applicability beyond two-dimensional materials. The MoS₂ and WSe₂ utilized in our study have the technological capability for large-area growth, thereby enabling the possible fabrication of large-area and multi-pixel device arrays. Hence, our research holds substantial significance in the domain of adaptive device.

Correction list in the revised manuscript:

1. In page 2, “their potential application in machine vision systems is hampered by a sluggish adaptation process, making integration challenging.” was revised to be “their potential application in machine vision systems is hampered by a sluggish adaptation process, making the combination of convolutional neural network with adaptive device challenging.”
2. In page 3, “an ultra-fast adaptive machine vision has been achieved by integrating convolutional neural networks with bionic avalanche transistor.” was revised to be “an ultra-fast adaptive machine vision has been achieved by combining convolutional neural networks with bionic avalanche transistor.”
3. In page 4, “Prof. Chai Yang's team proposed an innovative concept that can significantly reduce the requirements of circuits and algorithms via integrating a two-dimensional (2D) bionic vision sensor with visual adaptation capabilities and a convolutional neural network.” was revised to be “Prof. Chai Yang's team proposed an innovative concept that can significantly reduce the requirements of circuits and algorithms via combining a two-dimensional (2D) bionic vision sensor with visual adaptation capabilities and a convolutional neural network.”
4. In page 5, “Although, 2D bionic vision sensors have been well developed with bio-inspired photopic and scotopic adaptation, but still encounter some issues or disadvantages, limiting their integration to machine vision.” was revised to be “Although, 2D bionic vision sensors have been well developed with bio-inspired photopic and scotopic adaptation, but still encounter some issues or disadvantages, limiting their combination with machine vision.”

5. In page 6, “Leveraging the microsecond-level rapid adaptation capability, the bionic avalanche transistor is further seamlessly integrated with a convolutional neural network.” was revised to be “Leveraging the microsecond-level rapid adaptation capability, the bionic avalanche transistor is further seamlessly combined with a convolutional neural network.”
6. In page 26, “Importantly, through the integration of convolutional neural networks with bionic avalanche transistor, an adaptative machine vision system has been achieved.” was revised to be “Importantly, through the combination of convolutional neural networks with bionic avalanche transistor, an adaptative machine vision system has been achieved.”

Question 2: The fundamental physical principle comes from the negative photoconductivity (NPC). As the core performances of all are shown in Fig. 5 d&e, the photocurrent increases with weakened light and decreases with stronger light. It has been widely reported that NPC is easily achieved in 2D material systems. But here the operation of the device requires applying 20 V. This is a disadvantage compared to other works presenting the NPC phenomenon. Hence, this type of device has very little potential to raise the interest of the electronics industry.

Answer 2:

We appreciate so much for your constructive comments. As the reviewer mentioned, the negative photoconductivity (NPC) in 2D material systems has been reported, and the induced mechanisms include defect capture/release of carriers, carrier transfer, and the photo-thermoelectric effect. To highlight the advantages of avalanche-induced NPC in this work, Table R1 consolidates the properties and parameters of NPC devices as reported. The advantages and difference of our device are reflected in the following aspects:

(1) **Spontaneous switching between NPC and PPC based on ambient light intensity.** In previous reports, the switching usually required manual configuration of gate voltage to modulate material defects, necessitating real-time monitoring of application scenarios. In this work, we have achieved the spontaneous transition

between NPC and PPC for the first time in 2D materials system through light-intensity-induced technology. This breakthrough benefits from the smooth transition between photoconductivity and avalanche effect. More importantly, this technique can be combined with the concept of sensing and computing integration, significantly expanding the potential applications of NPC in the field of biomimetic vision.

(2) **Retinal-like dynamic sensory responsivity and weak light detection capabilities.** Benefiting from the avalanche effect, the device achieves light detection at ultra-low light intensity (26.39 pW), reaching a high optical response of 7.6×10^4 A/W. More importantly, as the light stimulus shifts from dark to bright, the sensor's response undergoes significant changes in both magnitude and sign (from 7.6×10^4 to -1×10^3 A/W), mimicking the function of the transition between rod and cone photoreceptors in the retina under varying environmental light conditions. This feature endows our transistor with a unique biomimetic vision function which not possessed by previously reported NPC devices.

(3) **Ultra-fast photo-response on microsecond-level, four orders faster than reported NPC device.** In previous reported bionic visual device, the dependence on defect and interlayer carrier transport limits adaptation time to the order of seconds. As shown in Fig. 1, the feedforward inhibition is faster and more predictable than feedback inhibition employed in previous 2D NPC devices. Taking advantage of avalanche and feedforward inhibition, our device can possess ultra-fast visual adaptation of 108 and 268 μ s, respectively, that is far beyond reported NPC devices. This is of great importance for rapid response and emergency operation of machine vision systems upon the complex environment.

Overall, we introduced the avalanche effect for the first time, achieving spontaneous switching of positive and negative photoconductivity controlled by light intensity, with negative photoconductivity regulated by feedforward inhibition circuit. More crucially, the response speed of our device's positive and negative photoconductivity effects is orders of magnitude faster than other reported mechanisms, demonstrating our device's broad application potential beyond visual adaptation.

As for the large operation voltage as you concerned, we are particularly grateful to you for pointing out its lack of advantage compared to previously reported devices. It is initially high to ensure the device enter the avalanche region. As an avalanche phototransistor, our device shows the superior avalanche performance and lower breakdown voltage compared to the reported 2D avalanche transistors.

To further improve the performance and reduce the power consumption following the reviewer's suggestion, we have implemented the following optimizations to improve the performance and power consumption of our device: (1) The channel length, a critical parameter affecting the bias control over the channel carriers, has been substantially reduced from 4.3 to 0.8 μm . As illustrated in **Fig. R1**, the breakdown voltage dropped from **10.7 V to 5.48 V**, and the applied bias for adaptation decreased from **20 V to 7.5 V**. (2) The extension of the gate electrode enhanced the gate modulation capability of WSe₂ over the MoS₂ channel, effectively reducing the dark current (from ~nA to ~pA) and improving the on/off ratio. (3) Constructing the device under vacuum conditions has effectively mitigated the influence of interface states and defect states. In summary, the comparison in **Table R1** illustrates our device's significant advantages in adaptation time, mechanism innovation, and photoconductivity switching characteristics, enhancing the potential application value in the field of negative photoconductivity.

Fig. R1. **a**, Output characteristics at different gate voltage (V_{GS}) in logarithmic scale of y-axis. **b**, Output characteristics of the device under various laser power with $V_{GS} = -3$ V in logarithmic scale of y-axis. **c**, Normalized real-time dependent current with different bandwidth under $V_{DS} = 7.5$ V in background illumination of 125.62 and

2158.43 pW, respectively. The initial conditions for scotopic and photopic adaptation are set to strong light power (2158.43 pW) and darkness, respectively.

Table R1. Comparison of the proposed negative photoconductivity devices with previous reports.

Sensor structure	Response time	Photoconductivity	Operating Mechanism	Operating voltage	Switching condition	Circuit motif	Ref.
MoS ₂ /WS e ₂	108, 268 μs	Positive, Negative	Avalanche	V _{DS} = 7.5 V, V _{GS} = -3 V	Light intensity	Feedforward inhibition	This work
MoS ₂	120, 120 s	Positive, Negative	Trapping & De-trapping	V _{DS} = 1 V, V _{GS} = -3/6 V	Gate Voltage	Feedback inhibition	[1]
ReS ₂ /U6 N	3 s	Negative	Trapping & De-trapping	V _{DS} = 1 V, V _{GS} = 40 V		Feedback inhibition	[2]
CsPbBr ₃ / PDPP-TT	120, 120 s	Positive, Negative	Trapping & De-trapping	V _{DS} = -10 V, V _{GS} = -20 V	Light Time	Feedback inhibition	[3]
CsPbBr ₃ / MoS ₂	2 s	Negative	Trapping & De-trapping	V _{DS} = 0.1 V, V _{GS} = -20 V		Feedback inhibition	[4]
γ-InSe nanoflake	150 s	Negative	Photo- pyroelectric and Photo- thermoelectric	V _{DS} = 0 V, V _{GS} = 0 V		Feedback inhibition	[5]
InP /ITZO	300/300 s	Positive, Negative	Trapping & De-trapping	V _{DS} = 1 V, V _{GS} = -2/3 V	Gate Voltage	Feedback inhibition	[6]
Tellurium	20, 20 s	Positive, Negative	Trapping & De-trapping	V _{DS} = 1 V, V _{GS} = 0 V	Environmental gases	Feedforward inhibition	[7]
ReS ₂ /h- BN/MoS ₂	12, 30 s	Positive, Negative	Interlayer carrier transport	V _{DS} = 1 V, V _{GS} = - 60/60 V	Gate Voltage	Feedforward inhibition	[8]

Correction list in the revised manuscript:

1. In page 22, “Supplementary Table 1 summarizes reported negative photoconductive devices with potential applications in visual adaptation to highlight the advantages of our device.” has been added.

Correction list in the revised Supporting Information:

1. In page 14, “Table S1. Comparison of the proposed negative photoconductivity devices with previous report.” has been added.

Question 3: One of the main drawbacks of this study, which cannot be remedied in a revision, is that the image recognition simulation part of the machine vision can be completely separated from the device part. This is because the software models can be arbitrarily modelled to obtain theoretical configurations of the network, including the input image brightness conditions if the brightness conditions are affecting the network performance in some way. In other words, the machine vision computing function still relies on the software processing, and the perception accuracy is irrelevant to the device to a large extent.

Answer 3:

We sincerely apologize for any misunderstandings caused by our previous insufficiently detailed description of the relevance between adaptive device and the brightness recognition accuracy of Convolutional Neural Network (CNN). Now, we would like to present a detailed and comprehensive explanation in the following aspects:

(1) CNN inherently lacks the capability to recognize the brightness of images, a limitation that remains even with explicit brightness parameters included during training. To evaluate the ability of convolutional neural networks (CNN) to classify images according to different brightness and features, we have conducted 30 learning experiments using 60,000 images from the MNIST dataset. Notably, during the training process, we input image brightness as an explicit additional parameter into the network. However, our test results indicate that the CNN lacks the ability to

accurately classify image features under conditions of high brightness, even with the brightness parameter.

(2) Due to the inherent limitations of CNN algorithms, enhancing the recognize capability of CNNs to image brightness through optimizing network configurations presents a limited scope of improvement. In addition, for already deployed networks, brightness training cannot improve the recognition rate. Adjusting the image brightness using an adaptive device provides a non-invasive hardware-level strategy, which can increase the accuracy to 98% and is applicable to all deployed CNNs. Therefore, it is crucial to construct an image brightness adaptation system for CNNs through adaptive devices.

(3) As you proposed, it is indeed possible to adjust image brightness through additional software, however, this approach requires significant computational power to process real-time image luminance parameters. Over/underexposed environmental brightness significantly reduces image features, and the generated electrical signals require means such as histogram equalization models to enhance image contrast, necessitating substantial computational resources and leading to a computational bottleneck. In addition, a hardware of photodetector is essential for the machine vision system to collect the brightness information. In order to simplify the algorithm and improve the recognition rate, Professor Chai from the Hong Kong Polytechnic University proposed a machine vision system comprising luminance adaptive devices and convolutional neural networks (*Nat. Electron.* 2022, 5, 84). Within this system, the adaptive devices not only capture luminance parameters but also process it by mimicking the human retina's adaptive function for high-precision recognition by CNN. Therefore, bionic vision devices capable of detecting and adapting to changes

in image brightness play a crucial role in enhancing CNN accuracy and reducing computational demands.

In our work, the bionic vision device demonstrates its fast brightness detection and adaptation capability, which effectively improves the feature classification accuracy of overexposed images. In Figs. R2a-b, we select the number "7" as the test image feature, with the illumination provided by the laser and the brightness data obtained by current mapping. Benefiting from the adaptive function, the device quickly adjusts the image brightness data from overexposed/underexposed to near-standard, which significantly improves the classification accuracy of the CNN, demonstrating the advantages of combining bionic vision devices with CNN. Overall, we achieved ultra-fast and spontaneous visual adaptation behavior through the switchover between avalanche effects and photoconductivity, overcoming the limitations of reported bionic vision devices and advancing the development and potential applications in the field.

Fig. R2. Adaptive machine vision. a, b, Recognition rate of adaptive machine vision as a function of time for (a) scotopic and (b) photopic adaptation. The inset shows the results of the visual adaptation simulation for MNIST image "7".

Correction list in the revised manuscript:

1. In page 25, “Notably, during the training process, we input image brightness as an explicit additional parameter into the network.” has been added.
2. In page 25, “It's concerning that under the +20% brightness condition, the accuracy is only 83%, indicating a substantial number of recognition errors.” was revised to be “It's concerning that under the +20% brightness condition, the accuracy is only 83%,”

indicating that CNN are unable to improve the accuracy of classification features even though they have obtained brightness parameter.”

3. In page 25, “In Figs. 6e and 6f, we select the number "7" as the test image feature, with the illumination provided by the laser and the brightness data obtained by current mapping.” has been added.

4. In page 25, “Figs. 6e and 6f showcase the outstanding performance of adaptative machine vision during scotopic and photopic adaptation, respectively.” has been deleted.

Question 4: Technical question. The device operates under a high voltage bias to reach the ‘avalanche’ state. Due to its JFET structure, the gate leakage current is important for reliability and power consumption. In Fig. 2f, there is only a small range of gate currents. As the V_{DS} is up to 20 V, the gate bias voltage also suffers more than 20 V. It is necessary to present the corresponding range of gate leakage current for a further judgment of whether the avalanche effect is valid or not.

Answer 4:

We deeply appreciate your valuable comments, which have significantly enhanced the quality of our manuscript. As you highlighted, the leakage current is crucial for the reliability and power consumption of JFET devices. **Fig. R3** illustrates the device leakage current as a function of bias under dark and illumination. In dark (**Fig. R3a**), the built-in electric field at interface between WSe₂ and MoS₂ effectively isolates the MoS₂ channel carriers from leaking into WSe₂, even at $V_{GS} = 0$ V and $V_{DS} = 10$ V. The drain current ($\sim\mu\text{A}$) is much higher than the leakage current ($\sim\text{nA}$) by more than 10^3 , confirming the validity of the avalanche effect and the high reliability of the device. Under illumination (**Fig. R3b**), the leakage current slightly increases with increasing light intensity due to the photoconductivity on the WSe₂. Nevertheless, the leakage current remains below 20 nA, which is still negligible compared to the drain current (tens of μA) under light illumination, underscoring the high reliability of our device under light conditions and the accuracy of the mechanism analysis in our manuscript. **Following your suggestion, we have included the leakage current as a**

function of bias under dark and illumination in the revised Supplementary Information.

Fig. R3. The leakage current as a function of bias under (a) dark and (b) illumination.

Correction list in the revised manuscript:

1. In page 16, “Notably, the drain current is higher than the leakage current by more than 10^3 , confirming the validity of the avalanche effect and the high reliability of the device, as shown in Supplementary Fig. 9.” has been added.

Correction list in the revised Supporting Information:

1. In page 6, “**Fig. 9.** The leakage current as a function of bias under (a) dark and (b) illumination.” has been added.

Reviewer #2:

Comment:

This manuscript demonstrated a sense-computing integrated junction-field-effect transistor (JFET) that can be used to emulate the visual adaptation beyond retina. The photodetector based on MoS₂/WSe₂ vdW heterostructure exhibits an extraordinary avalanche performance with low breakdown voltage of ~10 V and high multiplication

factor of 8.2×10^3 . In addition, the responsivity of the photodetector presents great changes in both magnitude and sign due to spontaneous transition of the photo-sensing mechanism between avalanche and photoconductive effect. Besides, the device shows ultra-fast adaptation process of 142 and 427 μs under simulated scotopic and photopic adaptation conditions, respectively. Furthermore, an ultra-fast adaptative machine vision has been achieved with respect to its remarkable microsecond-level rapid adaptation capabilities and robust image recognition with over 98% precision in both dim and bright condition. These results are interesting and the paper was well organized. Therefore, it can be published after the following major revisions:

Response:

We really appreciate so much for your kind comment and valuable suggestions that have helped us to much improve the quality of this manuscript. Thanks so much for thinking highly of our work as “*interesting and well organized*”. We have considered all the questions and made the corresponding revisions according to your suggestions. To reduce the device power consumption and dark current, we have optimized the device structure, resulting in improved avalanche and photovoltaic performance while reducing power consumption. In this revised version, we replaced all raw data with optimized data. Thank you very much again. The point-by-point response can be found as follows.

Correction list in the revised manuscript:

5. In page 11, “**Fig. 3 | Avalanche properties and operation mechanism. a**, Output characteristics at different gate voltage (V_{GS}) in logarithmic scale of y-axis. **b**, The electrical breakdown voltage (V_{EB}) and Multiplication factor as a function of V_{GS} . **c**, V_{EB} and Multiplication factor of our device and previously reported 2D devices. **d**, S_n/I as a function of frequency for different bias voltages at $V_{GS} = -3$ V. S_n and I are the noise spectral density and the corresponding current, respectively. **e**, Logarithmic plot of $1 - 1/M$ versus V_{DS} in avalanche multiplication at different V_{GS} . **f**, Ionization rate mapping at different bias and gate voltage.” has been revised.

6. In page 15, “**Fig. 4 | Light intensity dependent avalanche and operation mechanism.** **a**, Output characteristics of the device under various light power with $V_{GS} = -3$ V in logarithmic scale of y-axis. **b**, Photocurrent and avalanche gain extracted from the output characteristics as function of light power at $V_{DS} = 7.5$ V. The hollow and solid patterns represent the device in positive and negative photoconductivity, respectively. **c**, Responsivity with $V_{DS} = V_{EB}$ and avalanche responsivity with $V_{DS} = 7.5$ V as function of light power. **d**, Comparison of avalanche gain and responsivity between our device and previously reported avalanche photodetector. **e**, Logarithmic plot of $1 - 1/M$ versus V_{DS} in avalanche multiplication under different light powers. The inset is the ionization index as function of light power. **f**, Ionization rate mapping at different bias and power.” has been revised.

7. In page 20, “**Fig. 5. Avalanche photoresponse properties.** **a**, Fall and rise response time with V_{DS} under light power of 125.62 pW. **b**, Normalized I_{DS} as function of time with different fixed V_{DS} under continuous 125.62 pW light illumination. **c**, Normalized NPC and PPC response bandwidths at $V_{DS} = 7.5$ V, where the vertical dashed lines represent the -3 dB bandwidth. **d**, **e**, Normalized real-time dependent current with different bandwidth under $V_{DS} = 7.5$ V in background illumination of (**d**) 125.62 and (**e**) 2158.43 pW, respectively. The initial conditions for scotopic and photopic adaptation are set to strong light power (2158.43 pW) and darkness, respectively.” has been revised.

8. In page 24, “**Fig. 6. Adaptive machine vision.** **e**, **f**, Recognition rate of adaptive machine vision as a function of time for (**e**) scotopic and (**f**) photopic

adaptation. The inset shows the results of the visual adaptation simulation for MNIST image "7".” has been revised.

Question 1: In Supplementary Fig. 2a, why is the ratio of Mo and S not close to 1:2? In addition, why are the curves in Supplementary Fig. 2 and Supplementary Fig. 3 not smooth?

Answer 1:

We sincerely appreciate your comprehensive feedback. You highlighted a slight deviation in the ratio of Mo to S from the ideal 1:2 in Supplementary Fig. 2. We acknowledge that this deviation might be due to minor damage caused during the cutting process with Focused Ion Beam (FIB), an essential technique for obtaining cross-sectional Transmission Electron Microscopy (TEM) images. To enhance the readability of our paper, we have re-conducted the Energy Dispersive X-ray (EDX) analysis and presented the updated results in **Fig. R4**.

Regarding the smoothness issue you mentioned in Supplementary Figs. 2 and 3, we understand this primarily results from the limitations of the measurement instruments concerning output data points. The instruments employed for the cross-sectional TEM and Kelvin Probe Force Microscopy (KPFM) measurements are the Oxford Cypher S AFM and JEM-2100F, respectively. During the measurement process, we utilized the maximum data points permitted by the instruments to optimize the results.

Fig. R3. EDX analysis of cross sectional MoS₂/WSe₂ interface, corresponding to MoS₂.

Correction list in the revised Supporting Information:

1. In page 2, “**Fig. 2.** EDX analysis of cross sectional MoS₂/WSe₂ interface, corresponding to (a) MoS₂ and (b) WSe₂, respectively.” has been revised.

Question 2: From Fig. 3a and Supplementary Fig. 5, it can be observed that the dark current of the fabricated JFET device is relatively large (~1 nA), and the I_{on}/I_{off} ratio is relatively low (~10³). What could be the reasons for this? Are there any methods to reduce dark current and improve the I_{on}/I_{off} ratio?

Answer 2:

We greatly appreciate your insightful comments that have significantly contributed to enhancing the quality of our manuscript. Dark current, indicative of the standby current scale, depends on the junction quality, channel thickness, and gate voltage control capability within the JFET structure. In the manuscript, the dark current in the saturated state is considered as a critical parameter, but its value still exceeds 1 nA at $V_{GS} = -2$ V. Due to the thin layers of MoS₂ and WSe₂ (3.93 nm and 3.04 nm, respectively), the elevated dark current may originate from the defect states at the contact interface and the weaker gate voltage control on the channel.

To address these issues, we have implemented the following optimizations: (1) Device fabrication under vacuum conditions to effectively mitigate the influence of interface and defect states. (2) Expansion of the gate electrode to enhance the gate modulation capability of WSe₂ over the MoS₂ channel, further reducing dark current and improving the on/off ratio. (3) Reduction of the channel length from 4.3 μm to 0.8 μm to lower the breakdown voltage, thereby diminishing the increase in dark current induced by high bias. **Fig. R5** illustrates the optimized output characteristics. The saturation dark current at $V_{GS} = -2$ V was only **72 pA**, which is 27 times lower than that before optimization. At $V_{GS} = -3$ V, it was even below **10 pA**. The breakdown voltage is also significantly decreased from 10.7 V to **5.48 V**. Overall, these

optimizations have led to notable improvements in dark current, on/off ratio, and avalanche characteristics of the device.

Fig. R5. Output characteristics at different gate voltage (V_{GS}) in logarithmic scale of y-axis.

Question 3: From the simulated results in Fig. 3g, it can be observed that as V_{DS} increases, the ionization rate in MoS_2 gradually distributes away from the $\text{WeS}_2/\text{MoS}_2$ interface. What is the reason for this? In addition, why does V_{DS} regulate the avalanche effect?

Answer 3:

We greatly appreciate your constructive comment. The applied bias causes a voltage gradient across the channel, with the potential increasing from the source to the drain side. In this manuscript, due to the voltage gradient within the channel, the higher potential at the drain side induces a thicker depletion region, causing the channel to form an unaligned depletion region vertically. With an increase in bias, the channel on the drain side is preferentially pinched off by the depletion layer, subsequently expanding to form a pinch-off area. It is important to note that carrier drift is primarily located in the area away from the built-in electric field. Under the high bias, carriers near the pinch-off area are accelerated to impact the lattice and ionize, generating avalanche carriers. Hence, the ionization rate within the pinch-off area is significantly higher than that at the $\text{MoS}_2/\text{WSe}_2$ interface. Here, the regulation

of avalanche effect by the bias primarily involves two factors: (1) Increasing carrier drift velocity. (2) Inducing a thicker depletion layer at the drain side through the voltage gradient. We have added these relevant discussions to the revised manuscript.

Correction list in the revised manuscript:

1. In page 13, “The applied bias causes a voltage gradient across the channel, with the potential increasing from the source to the drain side. In this work, due to the voltage gradient within the channel, the higher potential at the drain side induces a thicker depletion region, causing the channel to form an unaligned depletion region vertically.” has been added.

2. In page 14, “Here, the regulation of avalanche effect by the bias primarily involves: 1. Increasing carrier drift velocity. 2. Inducing a thicker depletion layer at the drain side through the voltage gradient.” has been added.

Question 4: Regarding the simulation results in Fig. 3h, the authors claim that the ionization intensity at drain-side is less modulated due to that V_{DS} far exceeds V_{GS} . It is recommended to provide a more detailed explanation.

Answer 4:

We sincerely appreciate your comment. Under various gate voltages, the drain side maintains a high ionization rate. This phenomenon indicates that the voltage gradient induced by bias plays a dominant role in modulating the ionization rate at the drain side. As the negative gate voltage increases, despite the increased depletion layer thickness, the ionization rate on the drain side does not further improve due to the channel's limitation on the number of carriers. Overall, while the gate voltage has a weak regulating effect on the drain side ionization rate, it can extend the size of the ionization region. We have added relevant discussions in the revised manuscript.

Correction list in the revised manuscript:

1. In page 14, “the ionization intensity at drain-side is less modulated due to that V_{DS} far exceeds V_{GS} ” was revised to be “Notably, at $V_{DS} = 0$ V, as the $-V_{GS}$ increases, the

depletion layer thickness increases in an aligned manner. Consequently, under the same V_{DS} , a larger $-V_{GS}$ results in quicker transition to the avalanche state and forms a larger pinch-off region. However, due to avalanche mainly occurs near the pinch-off area causing a higher ionization rate, the larger $-V_{GS}$ does not further increase the ionization rate at the drain side”.

Question 5: In Fig. 4a, when V_{DS} is 20 V, the dark current I_{DS} of the device exceeds 1 μA , which may result in a relatively high power consumption. How do the authors balance power consumption and photoelectric performance?

Answer 5:

We sincerely appreciate your valuable comment. In the context of bionic vision devices, the most critical photoelectric performance metrics are the adaptation time and responsivity. To achieve industrial-level response times, we employed a strategy of switching between avalanche and photoconductance, which significantly reduces the adaptation time by more than 10^3 compared to previous reports. Here, the higher bias voltage not only triggers the avalanche effect but is also crucial for modulating the adaptation time and sensitivity. To reduce the power consumption while maintaining the photoelectric performance, we further optimized the device structure based on the strategy mentioned in comment 2, especially by shortening the channel length. This adjustment substantially decreased the avalanche voltage (from 10.7 V to 5.48 V) and the voltage applied for visual adaptation (from 20 V to 7.5 V), as detailed in **Fig. R6** for photoelectric performance.

Regarding the reduction of the operating current during avalanche, it can be achieved by reducing the channel width. However, in practical applications, a current above 1 μA is the minimum standard for signal recognition. To recognize smaller currents would necessitate increasing the cost of the accompanying circuitry. We believe that appropriately reducing the channel width, considering the accompanying circuitry and the minimum required photoelectric performance, can further decrease the scale of the power consumption. In summary, reducing both the width and length of the channel is a strategy to lower power consumption, allowing us to balance

photoelectric parameters and power requirements based on the performance needs in practical applications.

Fig. R6. **a**, Output characteristics of the device under various laser power with $V_{GS} = -3$ V in logarithmic scale of y-axis. **b**, Photocurrent and avalanche gain extracted from the output characteristics as function of light power at $V_{DS} = 7.5$ V. The hollow and solid patterns represent the device in positive and negative photoconductivity, respectively. **c**, Responsivity with $V_{DS} = V_{EB}$ and avalanche responsivity with $V_{DS} = 7.5$ V as function of light power. **d**, Normalized real-time dependent current with different bandwidth under $V_{DS} = 7.5$ V in background illumination of 125.62 and 2158.43 pW, respectively. The initial conditions for scotopic and photopic adaptation are set to strong light power (2158.43 pW) and darkness, respectively.

Question 6: Can this adaptation strategy extended to dynamic motion (Nature Nanotechnology, 2023, 18, 882-888; Nature Electronics, 2023, 6, 870-878)?

Answer 6:

Extending fast adaptation strategies to dynamic motion presents a promising avenue. Dynamic motion can transfer "spatial" and "temporal" streaming information to processing units and integrate it within units. Currently, mainstream dynamic

motion can superimpose multiple temporal motion information onto a single image, capturing the entire motion trajectory outline. As reported in Nature Nanotechnology, 2023, 18, 882-888, Professor Yang's team achieved this by trapping and incompletely releasing captured charges through defects in MoS₂, which requires longer photoresponse times. In our work, due to the fast photoresponse, it is challenging for the device to retain motion trajectories in one image unless the entire motion process is less than 200 μ s. Hence, the quick and slow response speeds have their advantages in different fields, and selecting appropriate devices for the application's functionality is crucial. Regarding event-driven vision sensors, as detailed in Nature Electronics, 2023, 6, 870-878, their essence lies in generating positive and negative spike signals as light increases and decreases, respectively. The authors achieved this mechanism through a floating gate layer with ultra-low power consumption. In future research, by optimizing the power consumption of avalanche transistors, we aim to fabricate an array of event-driven visual sensors for dynamic monitoring under lower light conditions, thereby expanding the operating brightness range of the device.

Question 7: The responsivity of the photodetector is generally significantly influenced by the applied bias. typically, a higher bias leads to a larger responsivity, but it can also introduce higher dark current. For the comparison in Fig. 4d, it is suggested that the author should add bias information.

Answer 7:

We are immensely grateful for your suggestions, which have significantly enhanced the quality of our manuscript. The external electric field indeed plays a pivotal role in modulating the performance of photodetectors. Considering the diverse device structures and their associated electric fields, including both bias and gate voltages, we have incorporated these variables into our comparison to offer a more comprehensive reading experience. As demonstrated in **Fig. R7**, our device, benefiting from a short channel length, requires a substantially lower bias voltage to induce avalanche compared to other reported avalanche phototransistors, positioning it at a leading level in the field.

Fig. R7. Comparison of avalanche gain, responsivity and external electric field between our device and previously reported avalanche photodetector.

Correction list in the revised manuscript:

1. In page 14, “Comparison of avalanche gain and responsivity between our device and previously reported avalanche photodetector” was revised to be “Comparison of avalanche gain, responsivity and external electric field between our device and previously reported avalanche photodetector”.

Question 8: Is the photogenerated voltage claimed by the authors the voltage formed by the separation of photogenerated electron-hole pairs?

Answer 8:

We deeply appreciate your insightful comments. In the manuscript, the formation of photogenerated voltage is attributed to the separation of photogenerated carriers, which is in the opposite direction of the built-in electric field, and it can also be interpreted as a thinning of the built-in electric field due to an increase in the carrier concentration in the MoS₂ channel. In the JFET structure, the photovoltage promotes the thinning of the depletion layer, which is confirmed by the photocurrent mapping. [Adv. Funct. Mater. 2021, 31, 2106105.] Remarkably, this phenomenon played a key role in our work by prompting a switch between avalanche and photoconductivity effects and inducing a negative photoconductivity effect, leading to an ultrafast visual adaptation function.

Question 9: The descriptions of Figs. 5d and 5e in the paper seem to be written in reverse. It is recommended to check them carefully.

Answer 9:

We sincerely appreciate your constructive feedback. After a thorough review, we have verified the accuracy of the description sequence for Figs. 5d and 5e. The misunderstanding may have arisen from the conditional description of the environmental illuminance, specifically the phrase “the environmental illumination at initial state is switched from bright (2158.43 pW) to dim (125.62 pW) conditions”, which was intended to convey the trigger for dark adaptation in response to the shift from bright to dim illumination, hence the mention of “at initial state” was unnecessary and has been removed. To improve clarity and avoid any potential confusion, we have amended the annotation in Fig. 5d with "bright (2158.43 pW) to dim (125.62 pW)", clearly delineating the change in illumination levels. Similarly, we have applied the same adjustment to the Fig. 5e of photopic adaptation to maintain consistency and prevent misunderstandings.

Fig. R8. Avalanche photoresponse properties. **a, b,** Normalized real-time dependent current with different bandwidth under $V_{DS} = 7.5$ V in background illumination of **(a)** 125.62 and **(b)** 2158.43 pW, respectively. The initial conditions for scotopic and photopic adaptation are set to strong light power (2158.43 pW) and darkness, respectively.

Correction list in the revised manuscript:

1. In page 21, “the environmental illumination at initial state is switched from bright (2158.43 pW) to dim (125.62 pW) conditions” was revised to be “the environmental illumination is switched from bright (2158.43 pW) to dim (125.62 pW) conditions”.

2. In page 19, “**Fig. 5d, e**, Normalized real-time dependent current with different bandwidth under $V_{DS} = 7.5$ V in background illumination of **(d)** 125.62 and **(e)** 2158.43 pW, respectively. The initial conditions for scotopic and photopic adaptation are set to strong light power (2158.43 pW) and darkness, respectively.” has been revised.

REVIEWER COMMENTS

Reviewer #1 (Remarks to the Author):

The authors have revised the manuscript with some new experimental data. However, it does not reach the degree of technical complexity, scientific correctness and impact of other manuscripts published in this field in this journal. I also think that it does not provide an application-compatible solution or potential perspective analyses, and that it not discloses enough information to support machine vision. The main points on which I based this decision are:

1 – One of the most important drawbacks of this study, which is unacceptable, is the distortion of the basic facts and self-referencing. This is a big problem because the authors are claiming and trying to convince the readers that their prototype has the potential for advanced sensing computing. For example, in the Abstract they claimed: "...that is far beyond human retina function with long adaptation process up to few minutes.". This is an unreasonable and very misleading statement without any reliable reference. Does the human eye need to be minutes long to respond? Seriously? Further, the authors also claimed "CNN inherently lacks the capability to recognize the brightness of images", which goes beyond the conclusions that can be drawn from this study. In fact, any change in the image parameters will re-affect the recognition accuracy, such as brightness and contrast. But this doesn't mean that CNNs can't learn these features. Maybe to require additional open learning designs during training, rather than simply defining them in a closed dataset. This should be rigorously analyzed and compared from the mathematical/computer science perspective, not the author's own words. As it is also out of the scope of this study, it should focus more on the actual performance of the device and be even less self-defining, rather than overclaiming.

2 – The whole study is based on a JFET with MoS₂/WSe₂ vdW heterojunction. It is acceptable to consider it as a photodetector, but there is no computational function within this process. Does this photodetector implement the multiply-accumulate computation (MAC) or differential calculation as in CNN? The author's claimed sense-computing process does not have an equivalent formula or schematic representation in the text. In other words, there are no surprises or new functional findings in this work, both at the individual device level or at neural network computation. It's a fairly fast-responding photodetector. The authors may think that a vague description could let the reviewers think that it is technically very sophisticated. This is not the case.

Overall, there is no insight or possible mechanism for such a sense-computing claim. The authors did not present sufficient evidence to show that the MoS₂/WSe₂ vdW JFET could achieve the claimed hardware integration advantages.

Reviewer #3 (Remarks to the Author):

The author proposed a neuromorphic transistor with feedforward inhibition. They used the avalanche effect occurring at the interface of MoS₂ and WSe₂ to achieve ultra-sensitive detection and high gain, as well as ultrafast adaptation processes under different lighting conditions. These results are interesting and the paper is rich in content and data. Therefore, it can be published after the following revisions:

- 1.The relationship between the network and the device is not shown in Figure 6. Please explain in detail how the convolutional neural network is combined with the device. For example, how to use the optical adaptive device to make the recognition rate of this network quickly improve.
- 2.Can you explain why the saturation current increases with the decrease of V_{gs} after avalanche?
- 3.How stable is the device, such as whether the avalanche effect will affect the durability of the device?
- 4.It seems that the coordinates in Supplement Figure 5 (why is the horizontal coordinate as V_{ds}) and 9 (why is the ordinate coordinate as I_{ds}) are wrong.
- 5.Figure 4 is inconsistent with the statement in the paper, please check carefully.

Responses to Reviewers' Comments

We acknowledge the reviewers for carefully reading our manuscript entitled “Adaptative machine vision with microsecond-level accurate perception beyond human retina” (NCOMMS-24-01836B) and providing constructive comments on our work. In response to these comments and questions, we have made a corresponding revision. We hope these are acceptable to allow the manuscript to be published.

The corresponding amendments to meet the referees' concerns have been marked in the revised manuscript. Our replies to the reviewers' suggestions and the corresponding changes are listed below.

The following are details of our response to reviewers' comments:

Reviewer #1:

Comments:

The authors have revised the manuscript with some new experimental data. However, it does not reach the degree of technical complexity, scientific correctness and impact of other manuscripts published in this field in this journal. I also think that it does not provide an application-compatible solution or potential perspective analyses, and that it not discloses enough information to support machine vision. The main points on which I based this decision are:

Response:

We appreciate your insightful comments on our research. You highlighted inaccuracies in the statements related to “human retinal adaptation time”, “the potential of CNNs in brightness recognition”, and “sensor-computation integrated devices” in our manuscript. To address your concerns and improve readability, we reviewed the relevant studies and corrected the imprecise statements. Furthermore, we would like to reiterate the novelty and significance of our work compared to previous

studies, demonstrating that our work meets the high standards required by Nature Communications.

In 2022, Professor Chai from Hong Kong Polytechnic University proposed a 2D bionic visual sensor with visual adaptation capabilities and showed that the bionic device can significantly improve the brightness recognition rate of three-layer convolutional neural networks (CNNs)¹. However, most bionic visual sensors use a mechanism involving additional gate feedback inhibition to control charge capture and release, resulting in long visual adaptation times. In a recent Nature Communications report, researchers achieved visual adaptation by applying strain to elastic perovskite devices; however, this new mechanism also led to a long visual adaptation time up to 150 seconds². Therefore, it is crucial to explore new bionic visual mechanisms to optimize visual behavior and achieve fast, high-frequency visual adaptation, which is the key technical challenge that 2D bionic visual sensors currently need to overcome.

In this work, we first proposed a novel and groundbreaking avalanche-tuned feedforward inhibition mechanism in bionic 2D transistors to achieve ultra-fast, high-frequency visual adaptation, addressing the existing critical challenges in this field. Its adaptation speed reaches the microsecond level, over 10^4 times faster than the retina and the most advanced bionic visual sensors. Additionally, we performed TCAD simulations of electric field distribution and impact ionization rates to further validate the bias- and light-tuned avalanche effect. Importantly, by integrating convolutional neural networks with the bionic avalanche transistor, the image recognition accuracy of CNNs exceeds 98% under both dim and bright lighting conditions. Therefore, avalanche-tuned bio-inspired visual devices can avoid long visual adaptation periods by introducing more predictive and faster feedforward inhibition circuits. This opens up significant prospects for broad applications in the field of machine vision, providing new insights and designs for bio-inspired visual systems while avoiding over-reliance on complex circuits and algorithms.

We believe our work can make a conceptual advance in designing novel bio-inspired visual device with unique avalanche-tuned feedforward inhibition

mechanisms to achieve a microsecond-level visual adaptation capability, the groundbreaking findings can align with the high standard requirement of Nature Communications, and we will appreciate so much that the you can change your mind and reconsider our work for publication.

References:

1. Liao, F. *et al.* Bioinspired in-sensor visual adaptation for accurate perception. *Nat Electron* **5**, 84–91 (2022).
2. Wang, C. *et al.* Strain-insensitive viscoelastic perovskite film for intrinsically stretchable neuromorphic vision-adaptive transistors. *Nat Commun* **15**, 3123 (2024).

Question 1:

One of the most important drawbacks of this study, which is unacceptable, is the distortion of the basic facts and self-referencing. This is a big problem because the authors are claiming and trying to convince the readers that their prototype has the potential for advanced sensing computing. For example, in the Abstract they claimed: “...that is far beyond human retina function with long adaptation process up to few minutes.”. This is an unreasonable and very misleading statement without any reliable reference. Does the human eye need to be minutes long to respond? Seriously? Further, the authors also claimed “CNN inherently lacks the capability to recognize the brightness of images”, which goes beyond the conclusions that can be drawn from this study. In fact, any change in the image parameters will re-affect the recognition accuracy, such as brightness and contrast. But this doesn't mean that CNNs can't learn these features. Maybe to require additional open learning designs during training, rather than simply defining them in a closed dataset. This should be rigorously analyzed and compared from the mathematical/computer science perspective, not the author's own words. As it is also out of the scope of this study, it should focus more on the actual performance of the device and be even less self-defining, rather than overclaiming.

Answer 1:

As for the question of visual adaptation time. We sincerely appreciate your insightful comments, which aim to enhance the accuracy and clarity of our manuscript. It is important to note that in the manuscript, we emphasized that our device possesses a microsecond-level visual adaptation capability. When we discuss about that “human retina function with long adaptation process up to few minutes”, here the “few minutes” is referring to the visual adaptation time, not the dynamic response time of human retina. Actually, the dynamic response time of retina is on the millisecond level which is also slower than that of our device. We sincerely apologize for any confusion caused by our unclear statements regarding “the visual adaptation time of the human retina” and “the potential of CNNs to recognize image brightness”. We hope to clarify these issues to alleviate your concerns or misunderstandings.

According to previous studies, visual adaptation in the human retina primarily relies on inhibitory synapses in the negative feedback loop between cone and horizontal cells. The process of spike formation in bright conditions (light adaptation) and its disappearance in darkness (dark adaptation) takes several minutes and is regulated by neurotransmitters dopamine and glutamate^{1,2}. Furthermore, visual adaptation time is dynamically regulated by factors such as age, electronic stress, and the state of the central nervous system³⁻⁵. It is important to note that visual adaptation is also strongly correlated with changes in environmental brightness, which might have led to your misunderstanding based on everyday experiences of incomplete visual adaptation. In our manuscript, the visual adaptation time of our device is in the microsecond range. Considering the multiple factors affecting visual adaptation time, such as environment brightness and age, we have removed the description of retinal visual adaptation time of few minutes from our manuscript.

As for the question of brightness recognition by CNN. Previously, Professor Chai Yang's team proposed an innovative concept by integrating a two-dimensional (2D) bionic visual sensor with visual adaptation capability with convolutional neural networks⁶. This integration can significantly reduce the demands on circuitry and algorithms. In this manuscript, we experimentally demonstrated that even after

training CNNs on fixed datasets with varying brightness levels, the accuracy in recognizing image brightness remains low, and our device can greatly enhance brightness recognition accuracy within microseconds, as shown in Fig. R1. We acknowledge the potential for CNNs to improve the accuracy of overexposed and underexposed image recognition through unknown optimizations such as open learning designs. At your suggestion, we have reviewed the conclusions in the manuscript regarding CNNs' recognition of overexposed images to ensure that we did not imply our device is the only method to improve CNNs' recognition accuracy of overexposed images.

Fig. R1. Confusion matrix results for 10,000 image recognition trials without (a) and with (b) device visual adaptation for a 20% brightness increase.

Correction list in the revised manuscript:

1. In page 3, “that is far beyond human retina function with long adaptation process up to few minutes” was revised to be “that is far beyond human retina function with long adaptation process”.
2. In page 22, “Unlike the long adaptation process of retina up to few minutes, the current of our bionic device can reach saturation states within 108 and 268 μ s for scotopic and photopic adaptation conditions” was revised to be “Unlike the long adaptation process of retina, the current of our bionic device can reach saturation states within 108 and 268 μ s for scotopic and photopic adaptation conditions”.
3. In page 25, “It's concerning that under the +20% brightness condition, the accuracy is only 83%, indicating that CNN are unable to improve the accuracy of classification features even though they have obtained brightness parameter” was revised to be “It's

concerning that under the +20% brightness condition, the accuracy is only 83%, indicating that CNN are unable to improve the accuracy of classification features even though they have obtained brightness parameter on a fixed dataset”.

References:

1. KEKCHYEYEV, K. Expediting Visual Adaptation to Darkness. *Nature* **151**, 617–618 (1943).
2. Weiler, R., Kohler, K. & Janssen, U. Protein kinase C mediates transient spinule-type neurite outgrowth in the retina during light adaptation. *Proceedings of the National Academy of Sciences* **88**, 3603–3607 (1991).
3. STEVEN, D. M. Relation Between Dark Adaptation and Age. *Nature* **157**, 376–377 (1946).
4. KRAVKOV, S. V. & GALOTCHKINA, L. P. Electrotonus in Colour Vision. *Nature* **155**, 605–606 (1945).
5. Brown, S. P. & Masland, R. H. Spatial scale and cellular substrate of contrast adaptation by retinal ganglion cells. *Nature Neuroscience* **4**, 44–51 (2001).
6. Liao, F. *et al.* Bioinspired in-sensor visual adaptation for accurate perception. *Nat Electron* **5**, 84–91 (2022).

Question 2:

The whole study is based on a JFET with MoS₂/WSe₂ vdW heterojunction. It is acceptable to consider it as a photodetector, but there is no computational function within this process. Does this photodetector implement the multiply-accumulate computation (MAC) or differential calculation as in CNN? The author's claimed sense-computing process does not have an equivalent formula or schematic representation in the text. In other words, there are no surprises or new functional findings in this work, both at the individual device level or at neural network computation. It's a fairly fast-responding photodetector. The authors may think that a vague description could let the reviewers think that it is technically very sophisticated. This is not the case. Overall, there is no insight or possible mechanism for such a sense-computing claim. The authors did not present sufficient evidence to show that

the MoS₂/WSe₂ vdW JFET could achieve the claimed hardware integration advantages.

Answer 2:

We sincerely apologize for the ambiguity in our initial presentation affected your reading experience and led to a misunderstanding of the innovative aspects of our work. Here, we will revise our statements to clarify and provide a detailed analysis of the advantages of our work in terms of 2D bionic visual devices and image brightness recognition rate of convolutional neural networks (CNNs).

Compared to traditional photodetectors, our device switches between avalanche and photoconductive effects as the light intensity changes, thereby achieving the retinal visual adaptation function. This function involves both perception and signal processing, shifting part of the computational tasks from the central nervous system to sensory devices closer to the visual data source^{1,2}. These devices perceive brightness information and provide feedback regulation. By combining our bionic device with CNNs, the feature classification accuracy of overexposed and underexposed images through CNNs can be significantly improved. The detailed perception and signal processing process is as follows: the brightness data are obtained through our bionic device's current mapping, that is subsequently imported to CNN for processing. Thanks to the adaptive functionality, the device can quickly adjust the image brightness data from overexposure or underexposure to near-standard levels, significantly enhancing the classification accuracy of the CNN. This highlights the advantage of combining bionic vision devices with CNNs.

Therefore, in this manuscript, we described the prototype with visual adaptation capabilities as a sense-computing integrated device. However, as you pointed out, this does not align with traditional matrix calculations. To avoid misunderstandings among readers and reviewers, we have revised “sense-computing integrated” to “bionic visual” throughout the manuscript.

In 2022, Professor Chai developed a 2D visual sensor with visual adaptation capabilities that significantly improved the image brightness recognition rate of three-layer CNNs³. However, their visual adaptation time controlled by the feedback

inhibition circuit was too long (in the seconds range) due to its mechanism relying on defect capture and release. Additionally, the scotopic and photopic adaptation required manual configuration of positive and negative gate voltages. Our device, on the other hand, employs a novel mechanism of avalanche and photoconductive switching to achieve active visual adaptation at the microsecond level, with specific advantages including: 1. Automatic switching between scotopic and photopic adaptation behaviors based on ambient light intensity. 2. Fast dynamic light response and ultra-sensitive light detection capabilities beyond retina. 3. Ultra-fast visual adaptation process at the microsecond level, four orders of magnitude faster than reported 2D bionic visual devices, significantly reducing the time required for CNNs to recognize image brightness.

Correction list in the revised manuscript:

1. In page 3, “The sense-computing integrated junction-field-effect transistor (JFET) exhibits an extraordinary avalanche performance with low breakdown voltage (V_{EB}) of approximately 5.48 V and high multiplication factor of 5.29×10^5 ” was revised to be “The bionic visual junction-field-effect transistor (JFET) exhibits an extraordinary avalanche performance with low breakdown voltage (V_{EB}) of approximately 5.48 V and high multiplication factor of 5.29×10^5 ”.
2. In page 5, “To solve the above issues and introduce a distinctive working mechanism for visual adaptation, we developed a novel sense-computing integrated 2D transistor” was revised to be “To solve the above issues and introduce a distinctive working mechanism for visual adaptation, we developed a novel bionic visual 2D transistor”.
3. In page 9, “Fig. 2 | Characterization of sense-computing integrated device based on MoS₂/WSe₂ vdW heterostructure” was revised to be “Fig. 2 | Characterization of bionic visual device based on MoS₂/WSe₂ vdW heterostructure”.
4. In page 26, “We have fabricated a sense-computing integrated transistor that can be used to emulate the visual adaptation beyond retina” was revised to be “We have

fabricated a bionic visual transistor that can be used to emulate the visual adaptation beyond retina”.

References:

1. Wang, Z., Wan, T., Ma, S. & Chai, Y. Multidimensional vision sensors for information processing. *Nature Nanotechnology* (2024) doi:10.1038/s41565-024-01665-7.
2. Gollisch, T. & Meister, M. Eye Smarter than Scientists Believed: Neural Computations in Circuits of the Retina. *Neuron* **65**, 150–164 (2010).
3. Liao, F. *et al.* Bioinspired in-sensor visual adaptation for accurate perception. *Nat Electron* **5**, 84–91 (2022).

Reviewer #3:

Comment:

The author proposed a neuromorphic transistor with feedforward inhibition. They used the avalanche effect occurring at the interface of MoS₂ and WSe₂ to achieve ultra-sensitive detection and high gain, as well as ultrafast adaptation processes under different lighting conditions. These results are interesting and the paper is rich in content and data. Therefore, it can be published after the following revisions:

Response:

Thank you very much for your kind and valuable comments and suggestions, which have greatly improved the quality of our manuscript. We appreciate your high evaluation of our work, stating that “*These results are interesting and the paper is rich in content and data*”. In the last revision, we redesigned the device structure to reduce power consumption and updated optimized data. At lower bias voltages and non-linear avalanche gain limitation, the avalanche effect does not reduce the device's durability. Additionally, we included durability tests under periodic photopic and

scotopic adaptation cycles to address your concerns. We have considered all questions and made the corresponding revisions based on your suggestions. Thank you once again. Our point-by-point responses are as follows.

Question 1:

The relationship between the network and the device is not shown in Figure 6. Please explain in detail how the convolutional neural network is combined with the device. For example, how to use the optical adaptive device to make the recognition rate of this network quickly improve.

Answer 1:

Thank you very much for your constructive comments. To evaluate the ability of a convolutional neural network (CNN) to classify images based on different brightness and features, we performed 30 learning experiments using 60,000 images from the MNIST dataset. It is worth noting that we input image brightness as an explicit additional parameter to the network during training. However, our test results show that even with the luminance parameter, the CNN has low accuracy in accurately classifying image features under photopic and scotopic adaptation condition.

The bionic vision device demonstrates rapid brightness detection and adaptation capabilities, effectively improving the feature classification accuracy of overexposed and underexposed images. In Figures R2a and R2b, we selected the digit “7” as the test image feature. The photopic and scotopic adaptation conditions are simulated using lasers, and the brightness data are obtained through the bionic device’s current mapping subsequently imported to CNN for processing. Thanks to the adaptive functionality, the device can quickly adjust the image brightness data from overexposure or underexposure to near-standard levels, significantly enhancing the classification accuracy of the CNN. This highlights the advantage of combining bionic vision devices with CNNs.

In summary, we achieved ultra-fast and spontaneous visual adaptation behaviors through the switching between avalanche effect and photoconductivity, overcoming

the limitations of previously reported bionic vision devices. This advances the bionic vision field and its potential applications. To enhance the reading experience, we have added a detailed description of the measurement process in the manuscript.

Fig. R2. Adaptive machine vision. a, b, Recognition rate of adaptive machine vision as a function of time for (a) scotopic and (b) photopic adaptation. The inset shows the results of the visual adaptation simulation for MNIST image "7".

Correction list in the revised manuscript:

1. In page 25, “with the illumination provided by the laser and the brightness data obtained by current mapping” was revised to be “with the illumination provided by the laser and the brightness data obtained by current mapping subsequently imported to CNN for processing”.

Question 2:

Can you explain why the saturation current increases with the decrease of V_{GS} after avalanche?

Answer 2:

We deeply appreciate your insightful comments. Regarding the statement “decreasing V_{GS} can strength avalanche effect”, which might cause misunderstandings, we have revised it to “increasing $|V_{GS}|$ can strength the avalanche effect”. Figure R3 illustrates the effect of $|V_{GS}|$ on the avalanche effect. As $|V_{GS}|$ increases, the built-in electric field at $\text{MoS}_2/\text{WSe}_2$ increases, leading to a corresponding increase in the depletion layer thickness. Therefore, the distribution of the electric field and ionization rate are positively correlated with $|V_{GS}|$. Under the same V_{DS} , a larger $|V_{GS}|$

results in a faster transition to the avalanche state and forms a larger pinch-off (ionization impact) region, thereby enhancing the avalanche saturation current.

Fig. R3. Technology computer aided design (TCAD) simulated evolution of the channel electric field and ionization rate by varying V_{GS} .

Correction list in the revised manuscript:

1. In page 14, “The distribution of both electric field and ionization rate is negatively correlated with V_{GS} , indicating that decreasing V_{GS} can strength avalanche effect” was revised to be “The distribution of both electric field and ionization rate is positively correlated with $|V_{GS}|$, indicating that increasing $|V_{GS}|$ can strength avalanche effect”.

Question 3:

How stable is the device, such as whether the avalanche effect will affect the durability of the device?

Answer 3

We sincerely appreciate your valuable comments on our work. Durability is indeed a crucial parameter for avalanche photodetector (APD). Compared to traditional silicon-based APDs, the quantum confinement in van der Waals layered materials enhances Coulomb interactions, increasing the ionization rate during the impact ionization process¹. Thanks to the advantage of low-bias operation, the 2D APDs exhibit low-power consumption and high durability².

Notably, our device achieves lower power consumption through structural optimization (Figure 3c) compared to reported 2D APDs, exhibiting highest multiplication factor and very low breakdown voltage. As the laser power increases, the device gradually transitions from avalanche to photoconductive effect, which means that the avalanche gain does not linearly increase, preventing excessive carrier

accumulation that could affect the device's lifespan. To illustrate the durability more clearly, we tested the device for 500 cycles under both scotopic and photopic adaptation conditions, as shown in Figure R4. The results indicate that the avalanche effect does not affect the device's durability. We have added this durability test in revised manuscript.

Fig. R4. The device operates for 500 cycles under simulated scotopic (a-b) and photopic (c-d) adaptation conditions, where the data of Fig. R4 a and c are extracted from Fig. R4b and d.

Correction list in the revised manuscript:

1. In page 23, “More details about the frequency and light intensity dependent photo-response measurements are shown in Supplementary Figs. 16-20” was revised to be “More details about the frequency, light intensity and durability dependent photo-response measurements are shown in Supplementary Figs. 16-21”.

Correction list in the revised Supplementary Information:

1. In page 12, “Figure 21. The device operates for 500 cycles under simulated scotopic (a-b) and photopic (c-d) adaptation conditions, where a and c are extracted from b and d” has been added.

References:

1. Zhang, Z. *et al.* Approaching the Intrinsic Threshold Breakdown Voltage and Ultrahigh Gain in a Graphite/InSe Schottky Photodetector. *Advanced Materials* **34**, 2206196 (2022).
2. E. Fabris *et al.* Impact of Residual Carbon on Avalanche Voltage and Stability of Polarization-Induced Vertical GaN p-n Junction. *IEEE Transactions on Electron Devices* **67**, 3978–3982 (2020).

Question 4:

It seems that the coordinates in Supplement Figure 5 (why is the horizontal coordinate as V_{DS}) and 9 (why is the ordinate coordinate as I_{DS}) are wrong.

Answer 4:

We sincerely appreciate your comprehensive feedback and valuable suggestions, which have significantly improved the quality of our manuscript. As you correctly pointed out, there were errors in the coordinates of the Supplementary Figures. Specifically, Supplementary Fig. 5 displays the transfer curves of the device as a junction transistor, where the horizontal coordinate should be V_{GS} instead of V_{DS} . Additionally, Supplementary Fig. 9 shows the leakage current under bias, where the ordinate coordinate should be I_{GS} instead of I_{DS} . We have carefully reviewed and corrected all the figures throughout the manuscript to ensure accuracy and enhance the reading experience. Thank you again for your meticulous comments.

Correction list in the revised Supporting Information:

1. In page 4, “**Figure 5.** Transfer characteristics of the device acting JFET. The transfer curves at V_{DS} at 3, 4 and 5 V are nearly overlapped, showing the device enters the saturation region from 3 to 5 V” has been revised.

2. In page 6, “**Figure 9.** The leakage current as a function of bias under (a) dark and (b) illumination” has been revised.

Question 5:

Figure 4 is inconsistent with the statement in the paper, please check carefully.

Answer 5:

We sincerely appreciate your insightful comments, which have significantly contributed to improving the quality of our manuscript. During the last revision, we optimized the device structure and replaced the raw data with optimized data. Upon

careful review, we found that Figure 4 was not updated accordingly. We have now replaced Figure 4 with the updated optimized data. Thank you again for your attention to detail.

Correction list in the revised manuscript:

1. In page 15, “**Fig. 4 | Light intensity dependent avalanche and operation mechanism.**” has been revised.

REVIEWERS' COMMENTS

Reviewer #3 (Remarks to the Author):

The authors have well addressed my concerns. The work can be thus accepted.